# *Phox2b* mutation mediated by *Atoh1* expression impaired respiratory rhythm and ventilatory responses to hypoxia and hypercapnia

Caroline B Ferreira[1], Talita M Silva[2], Phelipe E Silva[2], Claudio L Castro[2], Catherine Czeisler[3], José J Otero[3], Ana C Takakura[1]\*, Thiago S Moreira[2]\*

[1]Department of Pharmacology, Instituto de Ciencias Biomedicas, Universidade de Sao Paulo, Sao Paulo, Brazil; [2]Department of Physiology and Biophysics, Instituto de Ciencias Biomedicas, University of Sao Paulo, Sao Paulo, Brazil; [3]Department of Pathology, College of Medicine, Ohio State University, Columbus, United States

\*For correspondence:
takakura@icb.usp.br (ACT);
tmoreira@icb.usp.br (TSM)

**Competing interest:** The authors declare that no competing interests exist.

**Abstract** Mutations in the transcription factor *Phox2b* cause congenital central hypoventilation syndrome (CCHS). The syndrome is characterized by hypoventilation and inability to regulate breathing to maintain adequate $O_2$ and $CO_2$ levels. The mechanism by which CCHS impact respiratory control is incompletely understood, and even less is known about the impact of the non-polyalanine repeat expansion mutations (NPARM) form. Our goal was to investigate the extent by which NPARM *Phox2b* mutation affect (a) respiratory rhythm; (b) ventilatory responses to hypercapnia (HCVR) and hypoxia (HVR); and (c) number of chemosensitive neurons in mice. We used a transgenic mouse line carrying a conditional *Phox2b*$^{\Delta 8}$ mutation (same found in humans with NPARM CCHS). We crossed them with *Atoh1*$^{cre}$ mice to introduce mutation in regions involved with respiratory function and central chemoreflex control. Ventilation was measured by plethysmograph during neonatal and adult life. In room air, mutation in neonates and adult did not greatly impact basal ventilation. However, *Phox2b*$^{\Delta 8}$, *Atoh1*$^{cre}$ increased breath irregularity in adults. The HVR and HCVR were impaired in neonates. The HVR, but not HCVR, was still partially compromised in adults. The mutation reduced the number of Phox2b$^+$/TH$^-$-expressing neurons as well as the number of fos-activated cells within the ventral parafacial region (also named retrotrapezoid nucleus [RTN] region) induced by hypercapnia. Our data indicates that *Phox2b*$^{\Delta 8}$ mutation in *Atoh1*-expressing cells impaired RTN neurons, as well as chemoreflex under hypoxia and hypercapnia specially early in life. This study provided new evidence for mechanisms related to NPARM form of CCHS neuropathology.

## Editor's evaluation

The present manuscript by Ferreira and colleagues is of potential interest to researchers working in the field of neural control of breathing and associated respiratory disorders. This study provides some novel insight into some genetic lesions that may underlie some developmental respiratory pathophysiologies.

## Introduction

Breathing is an essential physiological function soon after birth because it can rapidly regulate $O_2$ and $CO_2$ levels in the blood for the rest of our life. Oxygen is mainly sensed by peripheral chemoreceptors, while $CO_2$ is regulated by central chemoreceptors, and to a lesser extent by peripheral

chemoreceptors (*Smith et al., 2006*; *Nattie, 2011*; *Guyenet, 2014*; *Guyenet and Bayliss, 2015*; *Guyenet et al., 2019*).

Hypoventilation and inability to increase breathing under low oxygen and high $CO_2$ levels are one of the most impacting symptoms in patients with congenital central hypoventilation syndrome (CCHS). The paired like homeobox 2B (*Phox2b*) mutations are well known to be involved in the development of CCHS (*Weese-Mayer et al., 1993*; *Amiel et al., 2003*). CCHS-related *Phox2b* mutations occur in two major categories: a trinucleotide, polyalanine repeat expansion mutations (PARM) and a non-polyalanine repeat expansion mutations (NPARM), which includes missense, nonsense, and frameshift mutations (*Patwari et al., 2010*; *Ramanantsoa and Gallego, 2013*; *Moreira et al., 2016*). *Phox2b* NPARM deletions within exon 3 are correlated with severe CCHS phenotype with complete apnea, profound hypoventilation during sleep, and/or cause of post-neonatal infant mortality (*Amiel et al., 2003*; *Weese-Mayer et al., 2010*).

The mechanism by which CCHS impact respiratory control is incompletely understood. Thus, investigating how *Phox2b* mutation in specific neuronal population could contribute to better understand the clinical respiratory outcomes in CCHS. In a rodent experimental model, *Phox2b* PARM mutation is specific to retrotrapezoid nucleus (RTN), a well-known region involved with central chemoreflex control, impaired respiratory control, and ventilatory response to hypercapnia in neonates (*Ramanantsoa et al., 2011*). In contrast, hypoxic ventilatory responses are intact and potentiated (*Ramanantsoa et al., 2011*). Additionally, genetic deletion of *Phox2b* from atonal homolog 1 (*Atoh1*)-expressing cells, that include not only RTN neurons (peri VII region) but also neurons located in the intertrigeminal region (peri V region), also abolished ventilatory response to hypercapnia in neonates (*Ruffault et al., 2015*). The effect seems to be dependent on neuronal loss of ventral aspect of the parafacial region, also named RTN. Although peri V neurons might also be affected, resection of this region did not impact respiratory response to low pH levels in a brainstem preparation (*Ruffault et al., 2015*), suggesting that this region is not involved with central chemoreflex. However, the extent to which NPARM in regions that are involved with respiratory control and chemosensitivity remains an open question.

Recently, a human CCHS case postmortem proband was found and the mutation predictably causes a frameshift and a hypomorph protein (*Phox2b$^{Δ8}$*) (*Di Lascio et al., 2018*). The present mutation was used to generate a conditional transgenic mouse line that can be activated by cre recombinase and introduce the humanized NPARM *Phox2b$^{Δ8}$* mutation during different developmental phases and regions (*Nobuta et al., 2015*). Expression of NPARM *Phox2b$^{Δ8}$* mutation in the ventral visceral motor neuron domain (non-respiratory domain) induced apnea in newborns, loss of visceral motor neurons and *Phox2b* neurons in the RTN, and pre-Bötzinger complex dysfunction (*Alzate-Correa et al., 2021*). Thus, in the present study, we proposed to investigate the effect of NPARM *Phox2b$^{Δ8}$* mutation in regions that are directly involved with respiratory control and central chemoreflex. To achieve this we used *Atoh1$^{cre}$* line as a promoter. *Atoh1* is expressed during development in proliferating cells in the rhombic lip and in postmitotic neurons. In this independent site, postmitotic neurons are the only region that co-express *Phox2b* and *Atoh1* surrounding the paramotor neurons that involves facial motor nucleus (thus peri VII, parafacial/RTN neurons) and trigeminal motor nucleus (peri V). We proposed to investigate the effect of NPARM *Phox2b$^{Δ8}$* mutation in these regions on respiratory function, ventilatory chemoreflex to hypoxia and hypercapnia during neonatal and adulthood. In addition, we proposed to determine the effect of this mutation in the development of *Phox2b* chemosensitive neurons in the parafacial/RTN region. Our hypothesis is that NPARM *Phox2b$^{Δ8}$* mutation in Atoh1-expressing cells impairs respiratory control, ventilatory responses to hypoxia and hypercapnia, and parafacial/RTN chemosensitive neurons.

We found that NPARM *Phox2b$^{Δ8}$* in *Atoh1*-expressing cells suppressed breathing activity in response to hypoxia and hypercapnia in neonates. Surprisingly, it did not mainly affect baseline ventilation. We also showed that adult mutant mice increased irregular breathing pattern and the ventilatory response to hypoxia was partially compromised. While ventilatory response to hypercapnia completely recovered. Additionally, anatomical data showed reduced *Phox2b$^{+}$*/tyrosine hydroxylase (TH)$^{-}$ immunoreactivity and fos$^{+}$/TH$^{-}$-activated neurons by hypercapnia in the parafacial/RTN region. Together, our findings imply that NPARM *Phox2b$^{Δ8}$* in *Atoh1*-expressing cells affects the development of the parafacial/RTN chemosensitive neurons, and consequently impaired breathing under hypoxic and hypercapnic conditions especially in neonates. These data provided new evidence for mechanisms related to CCHS neuropathology.

## Results

### Functional respiratory changes observed in NPARM $Phox2b^{\Delta 8}$ in $Atoh1^{cre}$-expressing cells

In the first set of experiment, we investigated whether a conditional mutation of $Phox2b^{\Delta 8}$ in $Atoh1$-expressing cells affects ventilation during neonatal and adult phase. Given that all $Phox2b^{\Delta 8}$, $Atoh1^{Cre}$ mice survived, respiratory parameters were examined between 1–3 and 30–45 post-natal days.

Body weight during neonatal phase was not different between mutation vs. control littermates (2.2±0.2 g vs. control: 2.3±0.2 g; p=0.731; t=0.348; N=8–10/group). In contrast, mutant mice showed a slightly reduction in body weight compared to controls during adulthood (15±0.7 g vs. control: 17±0.8 g; p=0.031; t=2.393; N=8/group).

The $Phox2b^{\Delta 8}$ mutation in $Atoh1$-expressing cells did not affect respiratory frequency during both neonatal and adult phase (neonate mutant: 179±18 vs. control: 165±11 bpm, p=0.479, t=0.723; adult mutant: 231±7 vs. control: 218±6 bpm, p=0.137, t=1.574; *Figure 1A*). However, $V_T$ was higher in neonate mutant mice vs. control (neonate mutant: 13±0.9 vs. control: 9±0.4 µL/g, p=0.0007, t=4.219, *Figure 1B*). As a result, $V_E$ was higher in neonate mutant compared to control (neonate mutant: 2373±353 vs. control: 1541±123 µL/min/g, p=0.0274, t=2.428; *Figure 1C*). On the other hand, there was no difference in $V_T$ (adult mutants: 15±2 µL/g vs. control: 12±1 µL/g, p=0.1084, t=1.715; *Figure 1B*) and $V_E$ (adult mutants: 3577±370 vs. control: 2705±263 µL/min/g, p=0.0755, t=1.920; *Figure 1C*) in adults.

$Phox2b^{\Delta 8}$, $Atoh1^{cre}$ mutation during neonatal phase did not affect inspiratory time ($T_I$) (neonate mutant: 0.11±0.01 vs. control: 0.120±0.01 s, p=0.734, t=0.345; *Figure 1D*), expiratory time ($T_E$) (neonate mutant: 0.28±0.04 s vs. control: 0.32±0.03 s, p=0.484, t=0.716; *Figure 1E*) and total cycle duration ($T_{TOT}$) (neonate mutant: 0.40±0.05 s vs. control: 0.44±0.04 s; p=0.516; t=0.663; *Figure 1F*) compared to their control littermates. However, during adult phase, mice carrying $Phox2b^{\Delta 8}$ mutation exhibited a reduction in $T_I$ (adult mutants: 0.082±0.003 s vs. control: 0.096±0.003 s; p=0.0071, t=3.147; *Figure 1D*), and an increase in $T_E$ (adult mutants: 0.21±0.006 s vs. control: 0.19±0.005 s; p=0.0345; t=2.342; *Figure 1E*) that did not affect $T_{TOT}$ (adult mutants: 0.29±0.007 s vs. control: 0.29±0.006 s; p=0.557, t=0.600; *Figure 1F*).

To test whether the increase in VT found in the mutant neonates might be an artifact of the whole-body plethysmograph system, in a subset of neonate, respiratory parameters were analyzed using head-out system (data not shown). Although $V_T$ was higher in neonate mutant mice compared to controls (9.4±0.31 vs. control: 8.6±0.4 µL/g, p=0.1143), it did not reach statistic difference due the small number per group (N=4).

To investigate whether changes in body weight and respiratory parameters might be related to changes in metabolic rate, we also measure oxygen consumption ($VO_2$) in neonate and adult mice. $VO_2$ and $V_E/VO_2$ did not differ between mutant and control littermates during both neonatal and adult phase (*Figure 1G–H*). These results suggest that changes in body weight and respiratory parameters cannot be explained by changes in baseline metabolic rate.

### NPARM $Phox2b^{\Delta 8}$ in $Atoh1$-expressing cells increased the number of apneas and breath irregularity during adult life

We next analyzed whether $Phox2b^{\Delta 8}$ mutation in $Atoh1$-expressing cells increases the number of apneas and breath irregularity during both neonatal and adult phase. As previously mentioned, the genetic strategy used by the present study is known to affect Phox2b neurons in the parafacial region, and these neurons have been proposed to participate as a generator of respiratory rhythm (*Huckstepp et al., 2016*; *Huckstepp et al., 2018*). Interestingly, there was no difference in the number of apneas in neonate mutant compared to control (neonate mutant: 7±0.8 vs. control: 5±0.6 apnea/min; p=0.155; t=1.491; *Figure 1I*). However, during adulthood the number of apneas in $Phox2b^{\Delta 8}$ mutation was higher compared to controls (adult mutants: 7±0.8 vs. control: 3±0.2 apnea/min p=0.0007; t=4.351; *Figure 1I*; *Figure 2*).

Breath-to-breath interval was also used as an indicative of breath irregularity. *Figures 3A and 4A* illustrate breathing recording at rest in controls and mutant mice during both neonate and adult phases, respectively. $Phox2b^{\Delta 8}$ mutation did not alter breath-to-breath interval in neonates (neonate mutant: 0.35±0.06 vs. control: 0.33±0.03; p=0.838; t=0.207). In contrast, breath-to-breath interval

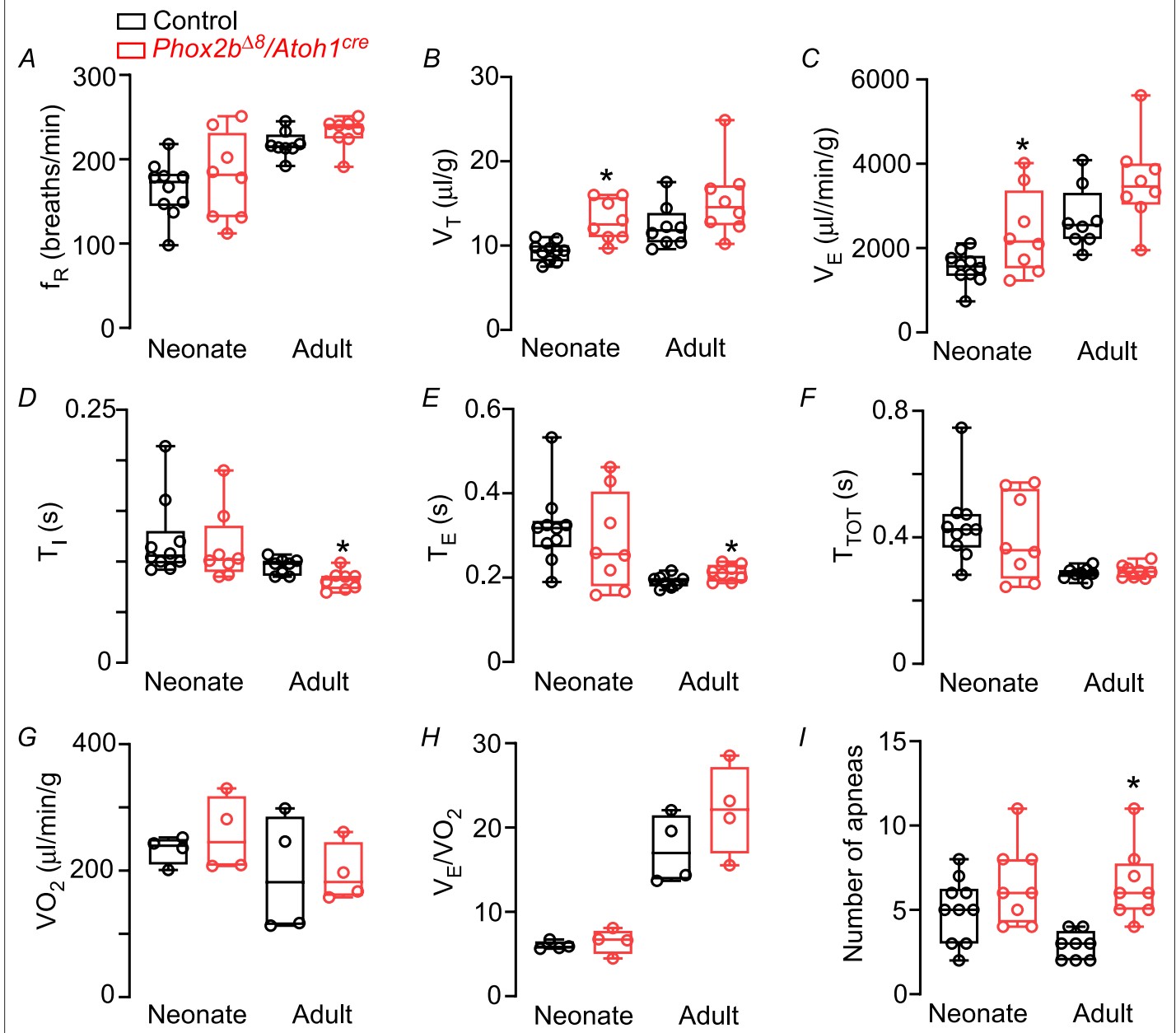

**Figure 1.** Functional respiratory changes observed in the *Phox2b^Δ8* mutation in *Atoh1^cre*-expressing cells. Changes in (**A**) respiratory frequency ($f_R$; breaths/min), (**B**) tidal volume ($V_T$; μL/g), (**C**) minute ventilation ($V_E$; μL/min/g), (**D**) inspiratory time ($T_I$; s), (**E**) expiratory time ($T_E$; s), (**F**) total cycle duration ($T_{TOT}$; s), (**G**) oxygen consumption ($VO_2$, μL/min/g), (**H**) air convection requirements $V_E/VO_2$ (a.u.), and (**I**) number of apneas in control and mutant (*Phox2b^Δ8*, *Atoh1^cre*) mice during neonatal and adult phase. Values are expressed as scatter dot plot with means ± SEM. Neonate (N=8–10/group); adult (N=8/group). *p<0.05 vs. control from Mann-Whitney U test.

The online version of this article includes the following source data for figure 1:

**Source data 1.** Raw respiratory parameters of control and Phox2bdelta8/Atoh1-cre mice.

was significantly higher in mutant adult mice compared to controls (adult mutants: 0.31±0.02 vs. control: 0.18±0.009; p<0.0001; t=5.505).

In addition to the time domain analysis, we also used a nonlinear method to investigate breath variability (*Li and Nattie, 2006*; *Patrone et al., 2018*; *Fernandes-Junior et al., 2018*). We quantified the distribution of the breath duration using the SD1 and SD2 parameters from the Poincare plots (*Figure 2A–B*). SD1 and SD2 were similar between mutant and control neonates (SD1: 132±31 ms vs. control: 159±36 ms, p=0.571, t=0.577; SD2: 135±28 ms vs. control: 223±54 ms, p=0.193, t=1.357)

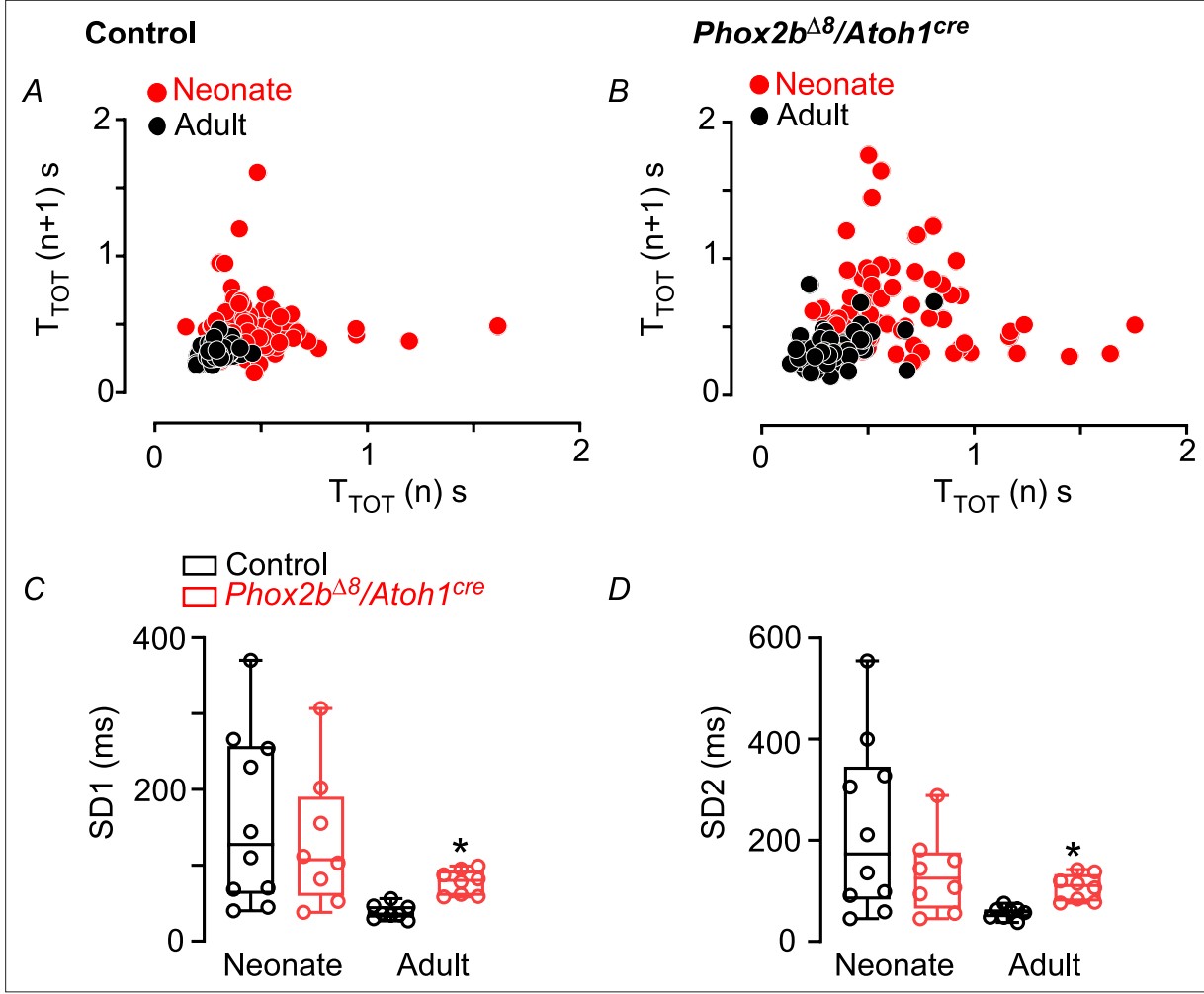

**Figure 2.** Breath variability increased in adult mice. Typical examples of Poincare plot graphs showing SD1 and SD2 from breath duration ($T_{TOT}$) vs. duration of the subsequent breath ($T_{TOT}$ n+1) in (**A**) control and (**B**) mutant mice ($Phox2b^{\Delta 8}$, $Atoh1^{cre}$) in neonatal (P3; red circles) and adult (P45; closed circles) phase. (**C**) Mean ± SEM of SD1 and (**D**) SD2 during neonatal and adult phases. Neonate (N=8–10/group); adult (N=8/group). *p<0.05 from Mann-Whitney U test.

The online version of this article includes the following source data for figure 2:

**Source data 1.** Raw breath variability of control and Phox2bdelta8/Atoh1-cre mice.

---

(*Figure 2C–D*). However, in agreement with the breath-to-breath interval, $Phox2b^{\Delta 8}$ mutation showed higher SD1 (77±6 vs. control: 38±3; p<0.0001; t=5.827) and SD2 (107±9 vs. control: 57±4; p=0.0002; t=4.995) in adult mice (*Figure 2C–D*). Altogether, these results suggest that breath irregularity is increased in adult mice carrying NPARM $Phox2b^{\Delta 8}$ mutation in $Atoh1$-expressing cells.

## NPARM $Phox2b^{\Delta 8}$ in $Atoh1$-expressing cells impaired ventilatory responses to hypoxia and hypercapnia in neonates

A common symptom experienced by patients with the CCHS is an impaired ventilatory response to hypoxia and hypercapnia (*Patwari et al., 2010*; *Moreira et al., 2016*). Therefore, we next explored whether a conditional $Phox2b^{\Delta 8}$ in $Atoh1$- expressing cells impairs ventilatory response to hypoxia and hypercapnia during the first days of life. *Figure 3A* illustrates examples of breathing recording at room air (left traces) and hypoxic challenge (middle traces) in a control (top) and mutant (bottom) mice 3 days after birth. We monitored baseline ventilation while neonates were breathing room air followed by 5 min of hypoxia. We analyzed the first minute of hypoxic exposure because longer than 5 min of low $O_2$ exposure is known to lower body temperature (*Kline et al., 1998*). *Figure 3B–D* illustrates percentage change in the respiratory frequency ($f_R$), tidal volume ($V_T$), and ventilation ($V_E$). As expected,

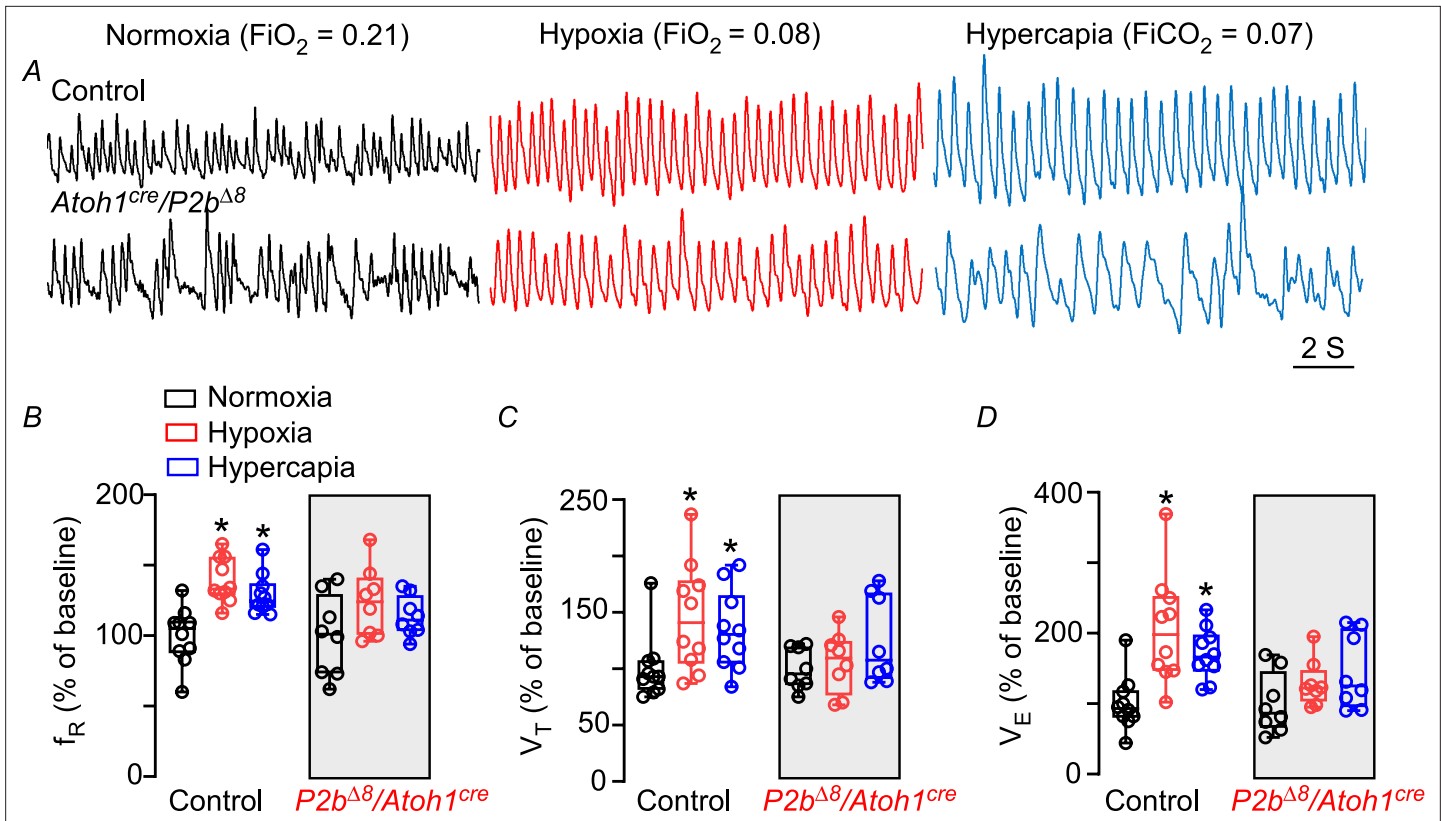

**Figure 3.** $Phox2b^{\Delta8}$ in $Atoh1^{cre}$ cells impaired ventilatory responses to hypoxia and hypercapnia in neonates. (**A**) Representative plethysmograph breathing traces in a control (top traces) and mutant ($Phox2b^{\Delta8}$, $Atoh1^{cre}$; bottom traces) neonate (P3) mice while ventilated with room air (normoxia; $FiO_2=0.21$); hypoxia ($FiO_2=0.08$); and hypercapnia ($FiCO_2=0.07$). Percentage changes produced by hypoxia or hypercapnia in neonate control and mutant mice. (**B**) Respiratory frequency ($f_R$; interaction: $F_{(2,32)}=0.8$, $p=0.455$; effect of mutation $F_{(1,16)}=4.3$, $p=0.052$; effect of hypoxia and hypercapnia: $F_{(2,32)}=10.5$, $p=0.0008$). (**C**) Tidal volume ($V_T$; interaction: $F_{(2,32)}=1.92$, $p=0.162$; effect of mutation $F_{(1,16)}=2.44$, $p=0.138$; effect of hypoxia and hypercapnia: $F_{(2,32)}=4.50$, $p=0.019$. (**D**) Minute ventilation ($V_E$; interaction: $F_{(2,32)}=3.32$, $p=0.048$; effect of mutation $F_{(1,16)}=4.48$, $p=0.0503$; effect of hypoxia and hypercapnia: $F_{(2,32)}=11.6$, $p=0.0002$). Values are expressed as scatter dot plot with means ± SEM. N=8–10/group. ANOVA two-way Dunnett's multiple comparisons test.

The online version of this article includes the following source data for figure 3:

**Source data 1.** Raw respiratory parameters of control and Phox2bdelta8/Atoh1-cre neonate mice under hypoxia and hypercapnia.

neonate control littermates increased $f_R \approx 40\%$ (from 100 ± 6% to 139 ± 5%; p=0.0016; ***Figure 3B***) and the $V_T$ increased 46% (from: 100±9% to 146 ± 15%; p<0.0001; ***Figure 3C***). That results in a significant increase in $V_E$ (from: 100±12% to 205 ± 24%; p<0.0001; ***Figure 3D***). In contrast, neonate mutant failed to significantly increase $V_E$ during hypoxia stimulus (from: 100±15% to 128 ± 11%; p=0.341; ***Figure 3D***). Note that we did not observe a significant change in both, $f_R$ (from: 100±10% to 123 ± 8%; p=0.269; ***Figure 3B***) and in $V_T$ responses (from: 100±10% to 105 ± 9%; p=0.910; ***Figure 3C***).

Our next goal was to investigate whether a conditional $Phox2b^{\Delta8}$ mutation impairs ventilatory response to hypercapnia. ***Figure 3A*** illustrates a typical respiratory trace from the same cre-negative and mutant neonate mice but now ventilated with 7% of $CO_2$ (right traces). As expected, neonatal control mice increased $f_R$ by approximately 30% during hypercapnia when compared to normoxia (from 100±6% to 129 ± 4%; p=0.0321; ***Figure 3B***). In addition, $V_T$ increased significantly from 100±9% to 134 ± 11% (p=0.042; ***Figure 3C***). Therefore, $V_E$ increased 71% in the control pups (from 100±12% to 171±11%; p=0.002; ***Figure 3D***). In contrast, $Phox2b^{\Delta8}$ mutation failed to significantly increase $V_E$ during hypercapnia (from 100±15% to 144±18%; p=0.096; ***Figure 3D***). The reduction was related to an impairment in $f_R$ (from 100±10% to 113±5%; p=0.360; ***Figure 3B***) and $V_T$ responses (from 100±6% to 124±13%; p=0.220; ***Figure 3C***).

These data suggest that $Phox2b^{\Delta8}$ in $Atoh1$-expressing cells affect ventilatory responses to hypoxia and hypercapnia during neonatal phase.

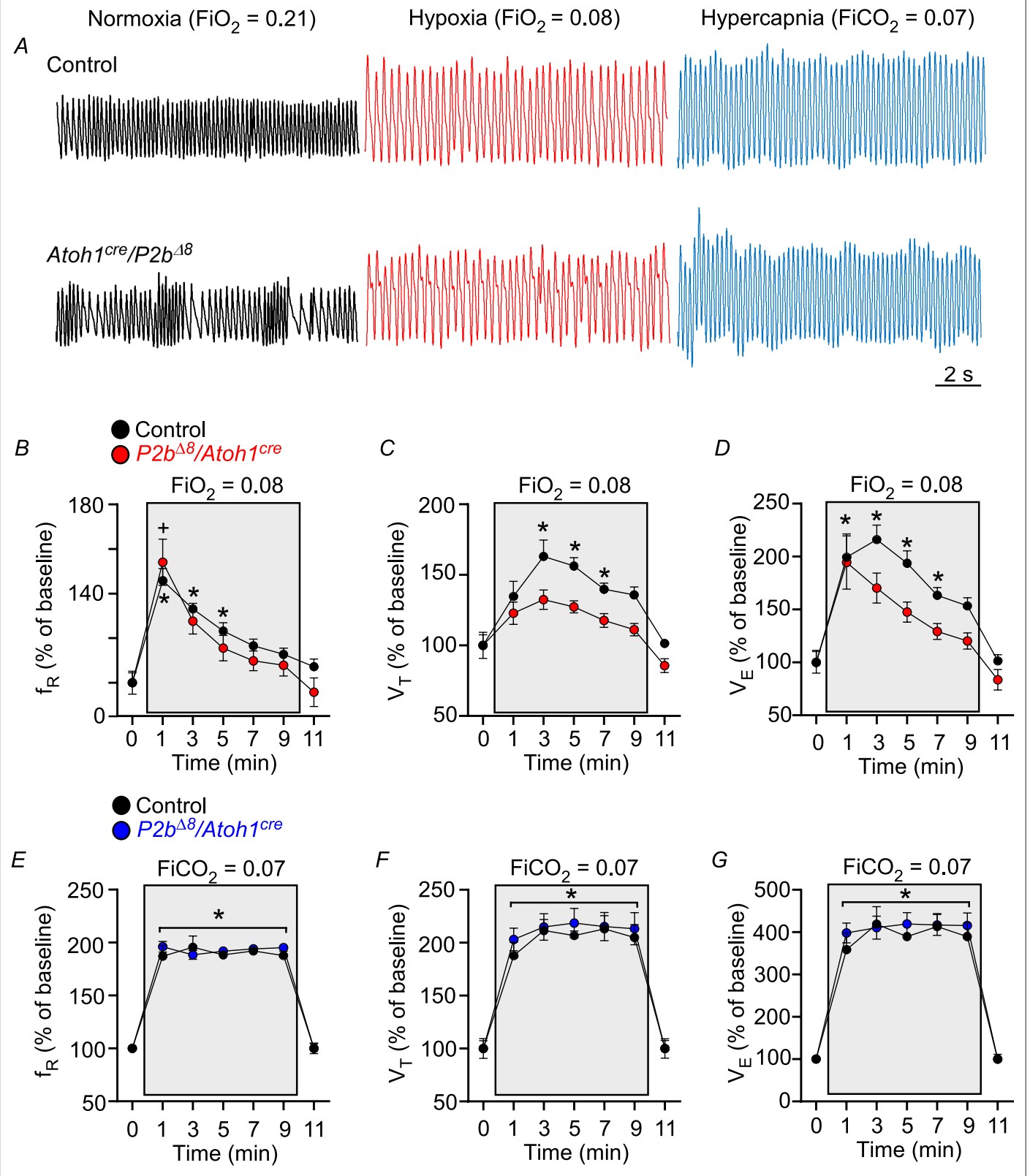

**Figure 4.** $Phox2b^{\Delta8}$ in $Atoh1^{cre}$ cells impaired ventilatory responses to hypoxia in adult. (**A**) Representative plethysmograph breathing traces in a control and mutant ($Phox2b^{\Delta8}$, $Atoh1^{cre}$) adult (P45) mice while ventilated with room air (normoxia; $FiO_2=0.21$); hypoxia ($FiO_2=0.08$); and hypercapnia ($FiCO_2=0.07$). Percentage changes produced by hypoxia or hypercapnia in adult control and mutant mice in (**B**) respiratory frequency ($f_R$; interaction: $F_{(6,84)} = 0.97$, $p=0.448$; effect of mutation $F_{(1,14)}=0.86$, $p=0.368$; effect of time of hypoxia: $F_{(6,84)} = 29.32$, $p<0.0001$; (**C**) tidal volume ($V_T$, interaction:

*Figure 4 continued on next page*

*Figure 4 continued*

F(6,84) = 1.26, p=0.285; effect of mutation F(1,14)=20.76, p=0.0004; effect of time of hypoxia: F(6,84) = 17.49, p<0.0001; (**D**) minute ventilation ($V_E$, interaction: F(6,84) = 1.20, p=0.316; effect of mutation F(1,14)=8.22, p=0.012; effect of time of hypoxia: F(6,84) = 23.92, p<0.0001). N=8/group. *p<0.05 vs. 21% $O_2$ in controls. +p < 0.05 vs. 21% $O_2$ in mutants. ANOVA two-way Dunnett's multiple comparisons test. (**E**) Respiratory frequency ($f_R$; interaction: F(6,84) = 0.56, p=0.763; effect of mutation F(1,14)=0.41, p=0.532; effect of time of hypercapnia: F(6,84) = 155.48, p<0.0001; (**F**) tidal volume ($V_T$, interaction: F(6,84) = 0.22, p=0.968; effect of mutation F(1,14)=0.31, p=0.585; effect of time of hypercapnia: F(6,84) = 69.77, p<0.0001); (**G**) minute ventilation ($V_E$, interaction: F (6,84)=0.34, p=0.914; effect of mutation F(1,14)=0.38, p=0.547; effect of time of hypercapnia: F(6,84) = 86.85, p<0.0001). N=8/group. *p<0.05 vs. 21% $O_2$ for both control and mutation group. ANOVA two-way Dunnett's multiple comparisons test.

The online version of this article includes the following source data for figure 4:

**Source data 1.** Raw respiratory parameters of control and Phox2bdelta8/Atoh1-cre adult mice under hypoxia and hypercapnia.

## Hypoxia, but not hypercapnic, ventilatory responses still partially compromised in the *Phox2b^Δ8*, *Atoh1^cre* adult mice

*Figure 4A* illustrates typical breathing traces in a control (top traces) and mutated adult mouse (bottom traces) while ventilated with room air (left traces), hypoxia (middle traces), and hypercapnia (right traces). *Figure 4B–D* illustrates changes in respiratory frequency, tidal volume, and minute ventilation before and during the hypoxic stimulus (10 min). As expected, breathing activity increased during hypoxia in control adult mice. Respiratory frequency increased at the first minute (from: 100±4% to 145±5%; p=0.0009), then slowly declined until the end of the hypoxia stimuli (*Figure 4B*). $V_T$ significantly increased from min 3 (from: 100±7% to 163±11%; p=0.029) and persistently elevated until min 7 (139±4%; p=0.027) (*Figure 4C*). Consequently, $V_E$ significantly increased from min 1 (from: 100±11% to 200±21%; p=0.024) to min 7 (163±7%; p=0.028) (*Figure 4D*).

In contrast, mutant mice only had a significant increase in $f_R$ at first minute of hypoxia from 100±5% to 154±10% (p=0.032; *Figure 4B*), but failed to increase $V_T$ across the stimulus (*Figure 4C*). Consequently, the increase in $V_E$ was compromised (*Figure 4D*). These results demonstrate that mutant mice had an impaired ventilatory response to hypoxia in the adult phase.

Interestingly, hypercapnia similarly increased $f_R$, $V_T$, and $V_E$ in both, mutant and control adult littermates (*Figure 4E–G*). These results demonstrate that mutant mice completely recovered the ventilatory response induced by hypercapnia in the adult phase.

## NPARM *Phox2b^Δ8* in *Atoh1*-expressing cells reduced *Phox2b* immunoreactivity in the parafacial/RTN region

The $CO_2$-sensitive cells of the ventral aspect of the respiratory parafacial/RTN region belong to a neuronal group with a well-defined phenotype characterized by the presence of *Phox2b* immunoreactivity and the absence of TH (henceforth called parafacial/RTN neurons) (*Stornetta et al., 2006*; *Shi et al., 2017*). According to prior evidence, *Phox2b* is predominantly expressed by the $CO_2$-activated neurons in the RTN region (*Stornetta et al., 2006*; *Shi et al., 2017*). But this marker is also present in a fraction of catecholaminergic neurons (known as C1) located close to the $CO_2$-sensitive neurons (*Stornetta et al., 2006*; *Shi et al., 2017*). The C1 neurons are normally bulbospinal blood pressure-regulating neurons (*Guyenet, 2006*) that can be distinguished from the $CO_2$-sensitive cells by the presence of TH (*Takakura et al., 2008*; *Takakura et al., 2014*; *Barna et al., 2012*; *Barna et al., 2014*). Thus, to assess the extent to which the mutation affects *Phox2b* expression in parafacial/RTN and C1 regions, we counted the number of *Phox2b*-expressing neurons that did not express TH (*Phox2b^+/TH^-*) and those that co-express TH (*Phox2b^+/TH^+*), respectively.

*Figure 5* shows typical photomicrographs and representative diagrams from several Bregma levels in a control (**A and B**) and mutant (**C and D**) adult mouse. The total number of *Phox2b^+* neurons (that include RTN and C1 neurons) was reduced in mutant adult mice (145±36 vs. control: 258±32; p=0.041; t=2.334; *Figure 5E*). The number of Phox2b^+/TH^+ (therefore C1 neurons) was similar between mutant and control mice (170±17 vs. control: 156±9; p=0.444; t=0.796; *Figure 5E*), which strongly suggests that *Phox2b* mutation in *Atoh1*-expressing cells did not compromise C1 neurons. On the other hand, the total number of *Phox2b^+/TH^-* neurons (RTN neurons) reduced ≈50% compared to controls (124±38 vs. control: 236±31; p=0.047; t=2.257; *Figure 5C, D and E*). These results indicated that *Phox2b* mutation in *Atoh1*-expressing cells compromised chemosensitive neurons (*Phox2b^+/TH^-*) in the parafacial/RTN region.

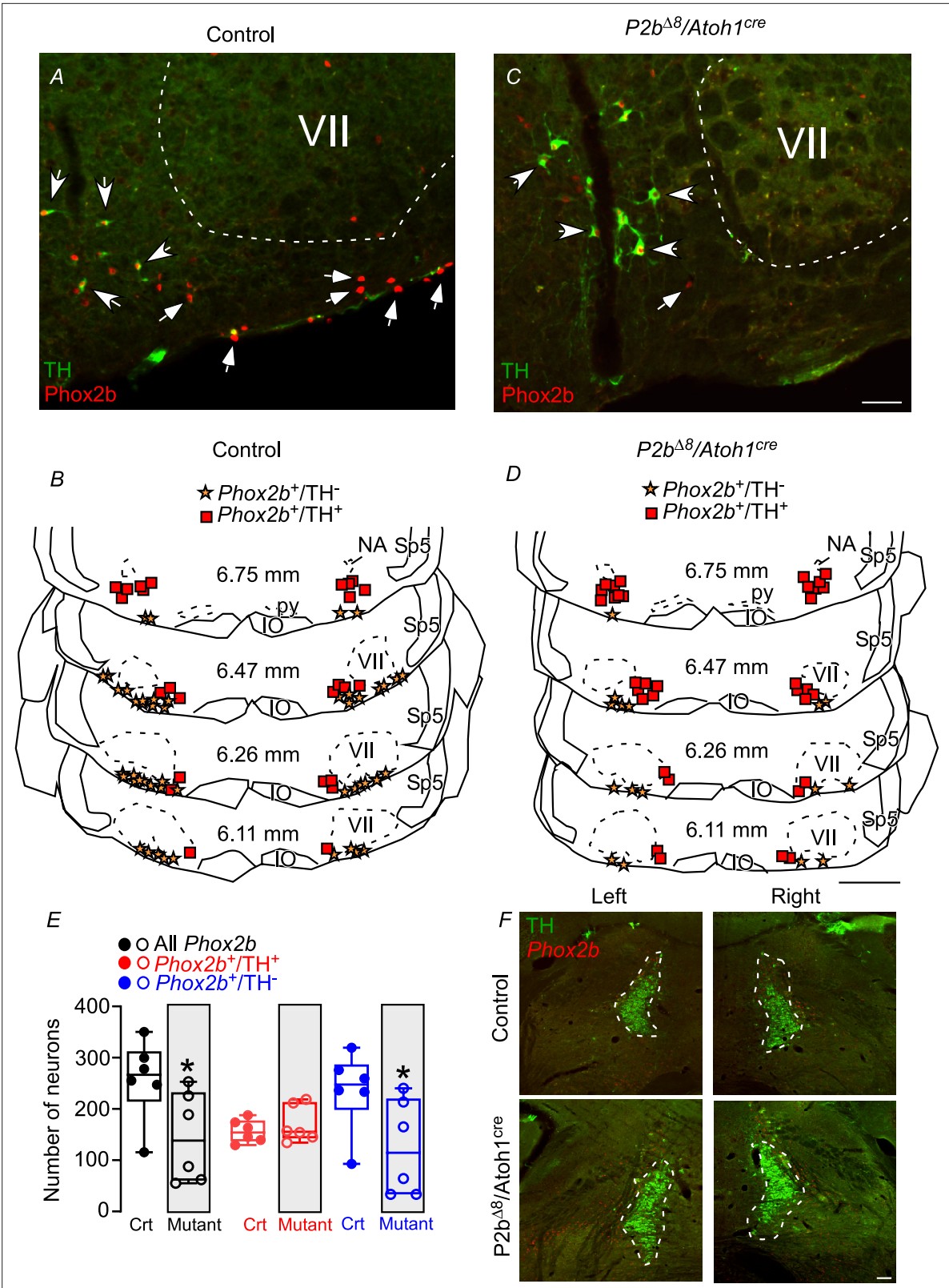

**Figure 5.** Adult mutant mice reduced *Phox2b* expression in the parafacial/retrotrapezoid nucleus (RTN) region. Photomicrographs of ventrolateral medulla from (**A**) control and (**C**) mutant (*Phox2b^Δ8^, Atoh1^cre^*) adult mice. Schematic drawings represent examples of coronal sections of ventrolateral medulla in (**B**) control and (**D**) *Phox2b^Δ8^, Atoh1^cre^* mutant mice. Each square represents immunoreactivity for *Phox2b* and tyrosine hydroxylase (*Phox2b^+^/ TH^+^*). The stars represent immunoreactivity for *Phox2b* and absence of TH (*Phox2b^+^/TH^-^*). The numbers in the middle of each section refer to the

*Figure 5 continued on next page*

*Figure 5 continued*

location caudal to the Bregma level (in mm) according to the Mouse Brain Atlas of *Franklin and Paxinos, 2015*. (**E**) Total number of cells that expressed *Phox2b* and TH immunoreactivity in the ventrolateral medulla (parafacial/RTN and C1 region) in control and *Phox2b$^{\Delta8}$, Atoh1$^{cre}$* (N=6/group). (**F**) Photomicrographs showing locus coeruleus and subcoeruleus region from control and mutant (*Phox2b$^{\Delta8}$, Atoh1$^{cre}$*) mice. *p<0.05 vs. control, unpaired t-test. Abbreviations: IO, inferior olive; NA, nucleus ambiguous; py, pyramid tract; Sp5, spinal trigeminal tract; VII, facial motor nucleus. Scale bar: C=50 µm applied to A; D=1 mm applied to B; F=100 µm.

The online version of this article includes the following source data for figure 5:

**Source data 1.** Raw numbers of neuronal profiles (Phox2b and TH) of control and Phox2bdelta8/Atoh1-cre adult mice.

We also examined the effect of the mutation on catecholaminergic cells located in the locus coeruleus (LC) (*Figure 5F*). Based on TH and *Phox2b* immunoreactivity, the mutation had no apparent effect on TH$^+$ neurons located in the LC region neither in *Phox2b$^+$* cells nor in the sub-LC region (*Figure 5F*).

## NPARM *Phox2b$^{\Delta8}$* in *Atoh1$^{cre}$* cells reduced the activation of ventral respiratory parafacial/RTN neurons by hypercapnia

As previously shown in *Figure 4*, *Phox2b$^{\Delta8}$, Atoh1$^{cre}$* in adult mice completely recovered ventilatory response induced by hypercapnia. However, these experiments did not rule out whether it involves activation of parafacial/RTN neurons. Thus, the next set of experiments were done to explore the involvement of the remaining parafacial neurons in response to hypercapnia. Mutant and control adult mice were challenged with hypercapnia and fos-immunoreactive was used as a reporter of cell activation. Hypercapnia is well known to induce fos expression in the rodent respiratory parafacial/RTN neurons (*Sato et al., 1992*; *Teppema et al., 1994*; *Fortuna et al., 2009*; *Kumar et al., 2015*; *Shi et al., 2021*). To differentiate between parafacial/RTN neurons and adjacent C1 neurons, we also analyzed the expression of TH. Therefore, parafacial/RTN neurons were defined by the presence of fos and absence of TH expression (TH$^-$) (*Stornetta et al., 2006*; *Barna et al., 2012*; *Barna et al., 2014*; *Shi et al., 2017*).

Excluding the facial motor nucleus, which expresses very low levels of fos-immunoreactive after hypercapnia, the ventrolateral medulla contains two clusters of fos-positive neurons centered predominantly within the rostral aspect. The fos-immunoreactive was expressed in both catecholaminergic (identified by TH$^+$) and non-catecholaminergic neurons (TH$^-$) in control and mutant mice (*Figure 6A and C*). In control animals, of the total 94±13 fos-immunoreactive neurons within the respiratory parafacial/RTN region, 86±12 (91%) were non-catecholaminergic, that is, presumably chemosensitive neurons (*Figure 6A, B and E*). On the other hand, in mutant mice, hypercapnia induced fos in only 47±7 neurons and a total of 37±8 were fos$^+$/TH$^-$ cells (reduction of 56%) (*Figure 6C, D and E*). These cells were generally located lateral to the TH$^+$ neurons and under the facial motor nucleus (*Figure 6C, D and E*). The neurons in this region are well known to belong to a cell group with a well-defined phenotype characterized by the presence of VGlut2 mRNA and the absence of both TH and choline acetyltransferase (*Stornetta et al., 2006*; *Shi et al., 2017*). In a subset of animals (N=3), fos expression was found in only six to eight neurons when exposed to room air (data not shown), which strongly suggested the effect of hypercapnia in activated neurons in the parafacial/RTN neurons.

These results indicate that *Phox2b$^{\Delta8}$* in *Atoh1*-expressing cells compromised the number of activated neurons in the parafacial/RTN region induced by hypercapnia.

## Discussion

In the present study, we used a conditionally activated NPARM patient-specific transgenic mouse model to investigate the effect of the mutant protein in *Atoh1*-expressing cells on respiratory function during neonatal and adult life. We found that the mutation resulted in (a) impaired hypoxic and hypercapnic ventilatory responses in neonates; (b) the ventilatory response to hypoxia, but not to hypercapnia, was reduced in adults; (c) the number of irregular breathing pattern increased in adults; (d) *Phox2b* expression within parafacial/RTN region (*Phox2b$^+$/TH$^-$*) reduced ≈50%; (e) no significant change in the number of catecholaminergic cells (TH$^+$) located in the ventrolateral medulla (C1 region) or in the dorsolateral pons (LC and subcoeruleus region); (f) the mutation also reduced the number of hypercapnic fos-activated neurons in the parafacial/RTN (fos$^+$/TH$^-$) by 56%. These findings demonstrate for the first time that NPARM *Phox2b$^{\Delta8}$* in *Atoh1*-expressing cells (in the parafacial/RTN and

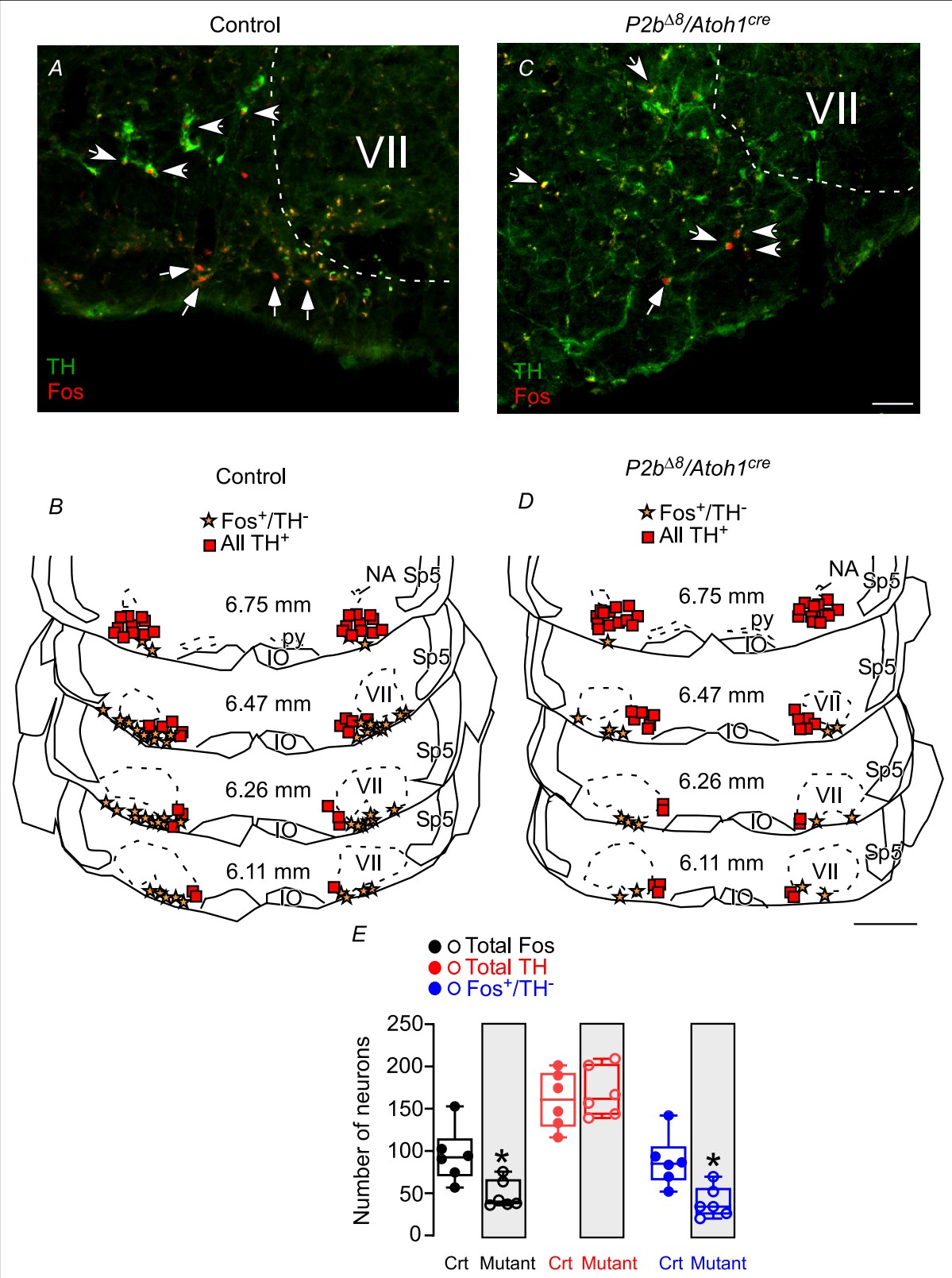

**Figure 6.** Fos-activated neurons in the parafacial/retrotrapezoid nucleus (RTN) region in response to hypercapnia are reduced in mutant mice. Photomicrographs of ventrolateral medulla from (**A**) control and (**C**) mutant (Phox2b^Δ8, Atoh1^cre) mice exposed to hypercapnia (FiCO2=0.07). Schematic drawings represent coronal sections of ventrolateral medulla in (**B**) control and (**D**) Phox2b^Δ8, Atoh1^cre mutant mice. Each square represents tyrosine hydroxylase immunoreactivity (TH+). The stars represent fos and the absence of TH (fos+/TH-). The numbers in the middle of the sections refer to the

*Figure 6 continued on next page*

*Figure 6 continued*

location caudal to the Bregma level (in mm) according to the Mouse Brain Atlas of *Franklin and Paxinos, 2015*. (**E**) Total number of cells that expressed fos and TH immunoreactivity in the ventrolateral medulla (respiratory parafacial/RTN region) in control and *Phox2b*$^{\Delta 8}$, *Atoh1*$^{cre}$ mice (N=6/group). *p<0.05 vs. control; unpaired t-test. Abbreviations: IO, inferior olive; NA, nucleus ambiguous; py, pyramid tract; Sp5, spinal trigeminal tract; VII, facial motor nucleus. Scale bar: C=50 µm applied to A; D=1 mm applied to B.

The online version of this article includes the following source data for figure 6:

**Source data 1.** Raw numbers of neuronal profiles (fos and TH) of control and Phox2bdelta8/Atoh1-cre adult mice under hypercapnia.

intertrigeminal region) affects regulation of breathing and chemosensory respiratory control for both, hypercapnia and hypoxia, especially in neonates. Furthermore, it showed that despite an impaired RTN region, anatomically and functionally during adulthood, the system adapted and developed appropriate responses to hypercapnia, but not to hypoxia (*Figure 7*).

## The effect of *Phox2b*$^{\Delta 8}$, *Atoh1*$^{cre}$ on baseline respiratory function

Our first goal was to investigate the effect of the *Phox2b*$^{\Delta 8}$ mutation on respiratory control at rest. The mutant neonates showed a slightly increase in tidal volume and consequently in total ventilation compared to their control littermates. It was a surprise since hypoventilation is usually found in both humans and experimental model of CCHS (*Amiel et al., 2003*; *Ramanantsoa et al., 2011*; *Carroll et al., 2014*; *Hernandez-Miranda et al., 2018*). Even when the mutation occurs specific to RTN neurons, it was demonstrated by a previous study that applied PARM *Phox2b* (*Phox2b*$^{27Ala}$) using the *Egr2*$^{cre}$ (*Krox20*$^{cre}$) promoter (*Ramanantsoa et al., 2011*). However, the reduction in ventilation in their study was due to reduction in total cycle duration with no change in tidal volume.

To further investigate whether the result in our study could be a bias of the plethysmograph method used (whole-body), in a subset of neonates, we used a more accurate method, the head-out plethysmograph. Despite the small number sample (N=4/group) we found similar results, showing a tendency to slightly increase in tidal volume and minute ventilation in the mutant neonate group. It is important to mention that, despite the small increase in ventilation there was no difference in oxygen consumption between mutants and control mice, indicating no changes in metabolic rate induced by the mutation.

Additionally, the small increase in ventilation does not seem to be a consequence of the genetic approach used by us, which target not only ventral parafacial/RTN region (peri VII), but also inter-trigeminal neurons (peri V) that express both *Phox2b* and *Atoh1* (*Ruffault et al., 2015*). The result suggests a direct effect of NPARM *Phox2b*$^{\Delta 8}$ mutation used in our study. Because inactivation of Phox2b from Atoh1-expressing cells, that is, the same population target in our study, found reduction in the ventilation in mutant neonates (*Ruffault et al., 2015*) and contradict with our study. Unfortunately, we do not know from the former study if the reduction in ventilation was due to change in tidal volume and/or respiratory frequency.

Whether the change is a consequence of NPARM *Phox2b*$^{\Delta 8}$ mutation specific to peri VII and/or peri V region, we do not have a clear picture because we do not use any strategy to target one of those population. Although, there is no data in the literature showing the contribution of peri V *Phox2b* neurons to regulate tidal volume. When *Atoh1* neurons were specifically removed from peri V region, it reduced tidal volume and consequently minute ventilation in mice at 3 weeks of age (*van der Heijden and Zoghbi, 2018*). It is also important to highlight that all studies cited above measured ventilation in neonates using the whole-body plethysmograph method. Thus, more accurate methods to measure neonatal tidal volume as head-out plethysmograph are required in further studies.

## The effect of *Phox2b*$^{\Delta 8}$, *Atoh1*$^{cre}$ on breath irregularity

In a healthy system, some level of respiratory variability is expected to occur since it can be affected by several factors, for instance chemical drive, excitatory and inhibitory input from many sources (*Khoo, 2000*). In the present study, adults carrying the *Phox2b*$^{\Delta 8}$ in *Atoh1*-expressing cells showed higher number of apneas, inter-breath interval (IBI), and breath variability. However, mutant neonates did not differ from controls littermates. Curiously, *Phox2b*$^{\Delta 8}$ in the *Nkx2.2*-derived progenitor domains (visceral motor non-respiratory neurons) reported apneic phenotype at birth and abnormal respiratory pattern (*Alzate-Correa et al., 2021*). Additionally, the loss of *Phox2b* neurons in the RTN region impaired inspiratory rhythmogenesis from pre-BötC. Pre-BötC neurons are known to receive *Atoh1*-dependent

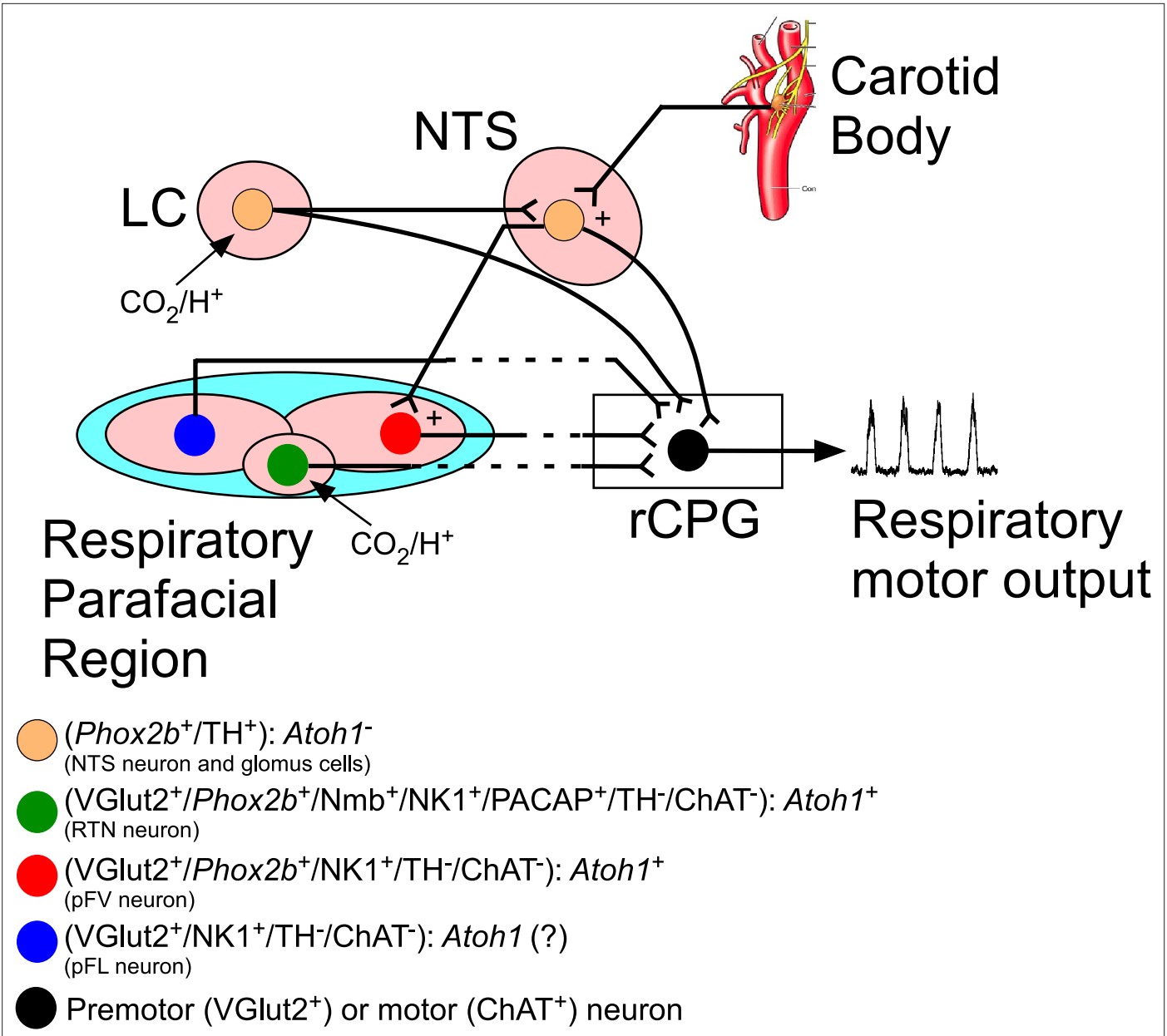

**Figure 7.** Schematic view of the mouse hindbrain control of breathing and the role of transcriptions factors and neuromodulators. The respiratory parafacial region (pF) contains neurons involved in breathing regulation. Within the ventral aspect of the pF, retrotrapezoid nucleus (RTN) could be defined as a cluster of neurons positive for *Phox2b*, neuromedin (Nmb), NK1, glutamatergic (VGlut2), pituitary adenylate cyclase-activating peptide (PACAP) and the absence of tyrosine hydroxylase (TH), choline acetyltransferase (ChAT), serotonin, GABA, and glycine. These neurons are activated by $CO_2$ via their intrinsic pH sensitivity and via inputs from the carotid bodies. The RTN of mice has a distinctive developmental lineage that relies on transcription factors *Egr2*, *Phox2b*, *Lbx1*, and *Atoh1*. *Phox2b* is the only one that remains expressed in adulthood. RTN progenitors originate from the dB2 domain of rhombomere 5. These progenitors are *Phox2b*-positive, switch on *Lbx1* at the postmitotic stage, migrate ventrally, and activate *Atoh-1* expression once they reach the region of the facial motor nucleus. In the respiratory pF also have distinct functional subgroup of neurons, that is, pF ventral neurons (non-RTN) and pF lateral neurons (expiratory oscillators). RTN neurons target various components of the respiratory central pattern generator (rCPG) and are presumed to play a key role in breathing automaticity during anesthesia, sleep, and quiet waking. The carotid body may also influence the activity of the rCPG neurons through connections that bypass the RTN (*Stornetta et al., 2006*; *Takakura et al., 2006*). The ventilatory response to $CO_2$ also has a contribution of the catecholaminergic neurons located in the locus coeruleus (LC). Here, we showed that breathing dysfunction of the humanized NPARM *Phox2b^{Δ8}* mutation in *Atoh1*-expressing cells is presumably mediated by loss of cells in the ventral parafacial region. Abbreviations: *Atoh1*, atonal homolog 1; ChAT, choline acetyltransferase; LC, locus coeruleus; Nmb, neuromedin B; NTS, nucleus of the solitary tract; NK1, tachykinin 1; PACAP, pituitary adenylate cyclase-activating peptide; *Phox2b*, paired like homeobox 2B; rCPG, respiratory central pattern generator; TH, tyrosine hydroxylase; VGlut2 (*Slc17a6*), vesicular glutamate transporter 2.

neuronal projections from both peri V and peri VII neurons (*Huang et al., 2012*). Therefore, $Phox2b^{\Delta 8}$ mutation in Atoh1-expressing cells could affect the excitatory tonic drive to pre-BötC neurons and increase irregular respiratory rhythm. Most studies applying different genetic strategies to target *Phox2b* neurons also reported higher number of apneas as soon as after birth (*Ramanantsoa et al., 2011*; *Ruffault et al., 2015*). We still do not know why higher irregular breathing patterns were only identified during adulthood, since breathing is well known to maturate post-natally.

## The role of $Phox2b^{\Delta 8}$, $Atoh1^{cre}$ on respiratory chemoreception

CCHS is characterized by impaired ventilatory response to hypoxia and hypercapnia. Our physiological data showed that both responses were blunted in neonates carrying NPARM $Phox2b^{\Delta 8}$ mutation in *Atoh1*-expressing cells. However, while hypercapnic ventilatory response completely recovered during adulthood, the hypoxic ventilatory response still partially compromised. The blunted response to hypercapnia in neonates is in line with other findings in the literature that used both PARM *Phox2b* mutation and genetically removed *Phox2b* from *Atoh1*-expressing cells in mice (*Ramanantsoa et al., 2011*; *Ruffault et al., 2015*). In addition, as previously found in adults carrying PARM *Phox2b* mutation restricted to RTN neurons, animals recovered hypercapnic response during adulthood (*Ramanantsoa et al., 2011*). Although similar findings were reported when deleting Atoh1 from peri V and peri VII region (*Huang et al., 2012*; *Ruffault et al., 2015*). It is unknown whether genetic deletion of *Phox2b* from *Atoh1*-expressing cells also recovers the ventilatory response to hypercapnia in adults.

The defect in the hypercapnic ventilatory response in neonate seems to be caused by anatomical and functional damage in neurons from peri VII region. In a brainstem spinal cord preparation, the increase in phrenic nerve activity in response to low pH was fully preserved after complete resection of peri V region (*Ruffault et al., 2015*). These in vitro data indicate that the peri V *Phox2b/Atoh1*-expressing neurons are not essential to chemosensitivity to $CO_2/H^+$. In addition, a recent study showed that loss of Atoh1 specifically from peri V *Phox2b/Atoh1* neurons did not compromise in vivo breathing responses to hypercapnia in neonates (*van der Heijden and Zoghbi, 2018*). Together, these results indicate that the compromised ventilatory response to hypercapnia is due to an impairment in parafacial/RTN neurons.

*Phox2b*-expressing RTN neurons located in the peri VII region are important $CO_2$ sensors in the brain and receive chemosensory inputs from other cells in the respiratory column in the brainstem (*Rosin et al., 2006*; *Guyenet et al., 2005*). Briefly, RTN neurons (a) are sensitive to small changes in $CO_2/H^+$ (*Mulkey et al., 2004*; *Onimaru et al., 2008*; *Wang et al., 2013*); (b) are in close opposition to numerous capillaries (*Onimaru et al., 2012*; *Hawkins et al., 2017*; *Cleary et al., 2020*), classifying these neurons to be critical to sense $CO_2/H^+$ in the blood; (c) receive afferents from many brainstem sites that contain putative chemosensors (*Rosin et al., 2006*); (d) respond with depolarization to activation of nearby acid-sensitive astrocytes (*Gourine et al., 2010*; *Wenker et al., 2010*; *Wenker et al., 2012*), and (e) receive excitatory connections from the carotid bodies (*Takakura et al., 2006*).

In the present study, NPARM $Phox2b^{\Delta 8}$ adult mice expressed only 50% of $Phox2b^+/TH^-$ (likely chemosensitive RTN/parafacial) neurons compared to their control. Neurons were located ventrally and laterally to facial motor nucleus. Similar to it, inactivation of *Phox2b* from *Atoh1*-expressing cells only expressed 40% of $Phox2b^+/Atoh1^+$ neurons from their controls in the RTN at 18.5 days of embryonic age. In addition, as in our study, cells were located ventral to facial nucleus. A massive loss of $Phox2b^+/TH^-$ neurons occurred when PARM *Phox2b* mutation was introduced in the RTN. Thus, loss of RTN chemosensitive neurons might be responsible for the blunted hypercapnic response in neonate. Therefore, the open question is what mechanisms enable the neonate mutant to maintain their ventilation, and presumably normal blood $PCO_2$ in a condition where the chemosensors neurons in the RTN were importantly reduced.

Here, we showed that although adult mutant mice recovered ventilatory response to hypercapnia, there was a reduction in both the $Phox2b^+/TH^-$ and $fos^+/TH^-$-activated neurons under hypercapnia in the parafacial/RTN region. Interestingly, former studies applied different strategies to manipulate *Phox2b* parafacial/RTN neurons, $CO_2$ response was only partially recovered in the adult life (*Ruffault et al., 2015*; *Ramanantsoa et al., 2011*; *Huang et al., 2012*; *Hernandez-Miranda et al., 2018*). Although, an extensive depletion of parafacial/RTN neurons occurred at embryonic ages. There is no information whether those neurons still depleted during adulthood and whether they are functional. Thus, it complicates further discussion with our finding. The recovery of the

CO$_2$ chemoreflex in adults in our study might be due to a late compensation of residual RTN neurons, peripheral chemoreceptor, and/or to some of the multiple chemosensors sites as previously described (**Nattie, 2011**).

One possibility is that carotid body compensates for the CO$_2$ drive to breathe and then through nucleus of the solitary tract (NTS) activates the respiratory column to maintain breathing activity. The plausible explanation emerges by considering that RTN neurons are strongly activated by carotid body stimulation and provide powerful excitatory input to the respiratory column (**Takakura et al., 2006**). They may thus be obligatory intermediates for relaying the CO$_2$ response when occur a loss of RTN neurons early in life. The second possibility is that RTN is not an obligatory site for central chemoreceptors in adults when the neurons are damaged at first days of life. Other candidates of chemoreceptor sites could assume the function. Those candidates are serotonergic neurons that have been reported to be pH-sensitive (**Wang and Richerson, 1999**; **Corcoran et al., 2009**), the noradrenergic neurons located in the LC (**Biancardi et al., 2008**) and glial cells (**Gourine et al., 2010**; **Wenker et al., 2010**; **Wenker et al., 2012**; **Sobrinho et al., 2014**).

The results of previous loss-of-function experiments to assess the role played by RTN neurons in the chemoreflex in adults are not entirely conclusive. In previous work, we evaluated the chemoreflex in which subsets of *Phox2b*-expressing neurons in the RTN were lesioned using toxin or pharmacological tools (**Takakura et al., 2006**; **Takakura et al., 2008**; **Takakura et al., 2013**; **Takakura et al., 2014**). Bilateral lesions of the neurokinin1 receptor-expressing neurons in the RTN region by injection of saporin conjugated to a substance P or injection of the GABA-A agonist muscimol reduced hypercapnic ventilatory response in adult rats (**Nattie and Li, 2002**; **Takakura et al., 2008**; **Takakura et al., 2013**; **Takakura et al., 2014**). However, these experiments lack specificity, and the extension of the lesion or inhibition is difficult to control. Using a more selective approach, **Marina et al., 2010**, applied a pharmacogenetic tool to silence RTN neurons. Rats that received injection of lentivirus vector expressing the allatostatin receptor from PRSx8 promoter reduced the hypercapnic ventilatory response after administration of allatostatin. However, the PRSx8 promoter used targets *Phox2a* and *Phox2b* neurons in the rostral aspect of the ventrolateral medulla, which includes RTN, C1 adrenergic, and A5 noradrenergic neurons (**Stornetta et al., 2006**; **Abbott et al., 2013**; **Burke et al., 2014**; **Malheiros-Lima et al., 2018**; **Malheiros-Lima et al., 2020**). Furthermore, it is important to mention that studies that tested loss of function of RTN/parafacial neurons in adult rodents need to be carefully discussed. Since it might exist important differences in neuronal plasticity, when comparing to neurons that were damaged early in the life.

Another important finding in our study was the compromised ventilatory response to hypoxia in both neonates and adult mutants. The hypoxic ventilatory response emerges from a physiological reflex of the already established notion that ventrolateral brainstem respiratory neurons are excited by peripheral chemoreceptors via a direct glutamatergic input from commissural NTS (**Guyenet, 2014**). Besides the di-synaptic excitatory pathway from commissural NTS to RVLM, we also know that we have a relay via the chemosensitive neurons of the RTN (secondary input) (**Takakura et al., 2006**). Thus, the compromised ventilatory response to hypoxia could be explained by the fact that this pathway was affected by the conditional *Phox2b$^{\Delta 8}$* mutation in *Atoh1*-expressing cells.

The impaired ventilatory response to hypoxia in neonates is in contrast with previous work that used PARM *Phox2b* mutation specific to RTN neurons. Interestingly, PARM *Phox2b* mutation in the RTN showed intact and even higher ventilatory response to hypoxia in neonates, despite the abrupt loss of RTN neurons (**Ramanantsoa et al., 2011**). This difference could be explained by the fact that in their study, peripheral chemoreceptors are potentialized in neonates. When neonate mutants were exposed to hyperoxia (100% O$_2$) they showed higher respiratory depression and apneas compared to their control. Although in the present study we have not tested hyperoxia in neonates, we found that at least in adults it did not cause any effect (data not shown).

The open question that needs to be investigated is by which mechanism NPARM *Phox2b$^{\Delta 8}$* mutation in *Atoh1*-expressing cells compromise chemosensory control of breathing in both neonates and adults. Such mechanisms may involve selective loss of neurons, disorganized respiratory circuits, that likely contributes to the irregular breathing pattern and apneic phenotype during adulthood. In addition, further studies could investigate whether the respiratory function and chemoreflex responses in mutants are altered during sleep stages.

## Conclusion

Our data established the NPARM $Phox2b^{\Delta 8}$ mutation in $Atoh1$-expressing cells with an impaired ventilatory response to hypercapnia and hypoxia in neonates. Although adult mutant mice recovered the ventilatory response to hypercapnia, the hypoxia ventilatory response still compromised, suggesting a reorganization within the chemoreflex pathways (*Figure 7*). In other words, the conditional $Phox2b^{\Delta 8}$ mutation in Atoh1-expressing cells affects the peripheral chemoreflex pathway and the important cells that serve as relevant chemosensors in the ventral aspect of the parafacial/RTN region. The remaining questions are: (a) How neonates were able to maintain their ventilation even with compromised hypoxic and hypercapnic ventilatory responses? (b) How the hypercapnic ventilatory response was restored in adult NPARM $Phox2b^{\Delta 8}$ mutation in Atoh1-expressing cells? Although parafacial/RTN neurons are particularly notable as they are important for respiratory chemoreceptors, substantial evidence has accrued supporting involvement of multiple cell types to maintain stable blood gases parameters, avoiding respiratory acidosis.

We showed that breathing dysfunction of the humanized NPARM $Phox2b^{\Delta 8}$ mutation in $Atoh1$-expressing cells is presumably mediated by loss of cells in the ventral parafacial region. Given that many other physiological processes could be affected by the mutation, our model may help to understand how specific brain areas and neurons generate and control complex behaviors more generally.

# Materials and methods

## Animals

This study was conducted in accordance with the University of Sao Paulo Institutional Animal Care and Use Committee guidelines (protocol number: 3618221019). Our goal was to introduce the NPARM in regions involved with respiratory function and chemoreflex. We used a transgenic mouse line with a cre-loxP-inducible humanized $Phox2b$ mutation defined as $Phox2b^{\Delta 8}$ and crossed them with $Atoh-1^{cre}$ mice (*Nobuta et al., 2015*; *Alzate-Correa et al., 2021*). These animals were bred with $Atoh1^{cre}$ mice to allow conditional expression of $Phox2b$ mutant gene in the parafacial and intertrigeminal region. Genotyping was verified by PCR (REDTaq ReadyMix # R4775, Sigma-Aldrich). The primers, genotyping details, and strain number of mice used are delineated in *Table 1*.

## Ventilation measurements

Breathing variables of neonatal (P1-3) and adult (P30-45) mice from both sexes were measured noninvasively in unanesthetized and unrestrained using the whole-body plethysmography closed system and the head-out pressure-plethysmography method (*Drorbaug and Fenn, 1955*; *Bartlett and Tenney, 1970*; *Mortola, 1984*; *Durand et al., 2004*; *Mortola and Frappell, 2013*; *Patrone et al., 2018*).

In neonates, part of the respiratory recording was done using the head-out pressure-plethysmography method (N=4/group) and part using whole-body plethysmography closed system (N=10–8/group).

The head-out pressure plethysmograph consists of separate head and body chambers that were 10 and 30 mL for P1-3 mice. The head and body chambers were separated by a pliable neck collar of plastic film that provided an air-tight seal between the two chambers. Three, premixed gas mixtures (room air 21% $O_2$, balance $N_2$; hypercapnia 7% $CO_2$, 21% $O_2$, balance $N_2$; hypoxia 8% $O_2$, balance $N_2$;

**Table 1.** Genotyping primers.

| Mouse line | Strain name | Strain # | Obtained from | Primers | Band sizes |
|---|---|---|---|---|---|
| $Atoh1^{Cre}$ | B6.Cg-Tg(Atoh1-cre)1Bfri/J | Jax: 011104 | Jackson Laboratories | Tg FWD 5'-CCG GCA GAG TTT ACA GAA GC-3' | Tg = 450 bp |
| | | | | Tg REV 5'-ATG TTT AGC TGG CCC AAA TG-3' | CTR = 324 bp |
| | | | | CTR FWD 5'-CTA GGC CAC AGA ATT GAA AGA TCT-3' | |
| | | | | CTR REV 5'-GTA GGT GGA AAT TCT AGC ATC ATC C-3' | |
| $Phox2b^{\Delta 8}$ | B6.129(Cg)-Phox2btm1Rth/J | Jax: 025436 | David Rowitch, UCSF | FWD 5'-GCC CAC AGT GCC TCT AA CTC-3' | Mutant = 450 bp |
| | | | | REV 5'-CGT ACT CTT AAA CGG GCG TCT C-3' | Wild type = 334 bp |

Oxylumen Gases Industriais Ltda, Sao Paulo, Brazil) were delivered continuously through the head chamber mice a flow rate of 40 mL/min for P1-3. The body chamber was sealed but had two ports, one for the differential pressure transducer (FE 141 Spirometer, ADInstruments, Sydney, Australia) used to monitor pressure oscillations associated with breathing and the other calibration port for injecting and withdrawing known volumes of gas (via a graduated syringe). Calibration of the system via injection of different volumes of air into the body chamber (0.2, 0.4, 0.6, 0.8 mL) established that the pressure signal (mV) was directly proportional to volume and that the relationship was linear ($R^2$=0.999). The pressure signal was amplified (FE 141 Spirometer, ADInstruments), digitized (200 Hz), and stored on computer via acquisition software (PowerLab system, ADInstruments/LabChart Software, version 7.3). The entire plethysmograph system was under a controlled temperature to maintain in the thermoneutral zone for P1-3 age between 32.5°C and 33.5°C (*Mortola, 1984*).

For whole-body plethysmography closed system, the plethysmograph chamber of neonate had 40 mL and was saturated with water vapor and thermoregulated at 32.5°C and 33.5°C (*Mortola, 1984*; *Durand et al., 2004*). The flow rate was set to 40 mL/min to avoid $CO_2$ and water accumulation. Breathing recording in adult mice was all done using whole-body plethysmography closed system in a larger chamber (500 mL) and flow rate was set to 500 mL/min. Experiments occurred at 24–26°C room temperature. The animal chamber was connected to a differential pressure transducer and to a preamplifier (FE 141 Spirometer, ADInstruments) to detect pressure oscillations when chamber was completely closed. Volume calibration was performed for each experiment by injecting 0.2–0.5 mL of air into the neonatal and adult chamber. The signal was digitalized using PowerLab system (ADInstruments). The sample rate was set as 1000 Hz and signal were filtered in 0.5–20 Hz bandwidth.

Breathing variables as breath duration ($T_{TOT}$; s), inspiratory time ($T_I$; s), expiratory time ($T_E$; s), tidal volume ($V_T$; μL/g), respiratory frequency (fR; breaths/min), and ventilation ($V_E$; μL/min/g) were analyzed offline using Lab Chart software (ADInstruments). Tidal volume in whole-body plethysmography was calculated as previously described (*Patrone et al., 2018*). Minute ventilation was defined by the product of breathing frequency and tidal volume. Breath variability was analyzed by IBI irregularity and it was defined as IBI irregularity = abs ($T_{TOT}$ (n+1) – $T_{TOT}$ (n))/ $T_{TOT}$ (n) (*van der Heijden and Zoghbi, 2018*). We also used a nonlinear method of analyses known as Poincare map. This method plots breath duration ($T_{TOT}$) vs. duration of the subsequent breath ($T_{TOT}$ n+1). We used a total of 100 breaths at rest condition. Next, we calculated SD1 and SD2 that describe the distribution of the points in the ellipse using the Kubios software (version 3.5.0) (*Brennan et al., 2002*). In summary, it was calculated the width of the variation perpendicular to (SD1) and along the line of identity (SD2) from the ellipse that describes the distribution of the points (*Brennan et al., 2002*).

To quantify breathing parameters, we first calculated the average of 30 s during a stable condition for each animal during normoxia, hypoxia, and hypercapnia. To quantify changes during hypoxia and hypercapnia, we normalized the data to baseline for each animal and then calculate the relative changes expressed as percentage. Spontaneous apnea-like events or respiratory pause was defined by the cessation of breathing greater than the average of one respiratory cycle to identify possible breathing pattern abnormalities. The duration of each apnea-like event was from the end of the first breath to the start of the following breath.

## Measurements of O$_2$ consumption

We used an $O_2$ analyzer (ADInstruments) that was connected to the output port of the animal's head chamber to pull air through the chamber at 100 mL/min for P1-2 and 500 mL/min for adult mice. A mass flow system (MFS, Sable Systems International, Las Vegas, NV, USA) was coupled to the outlet of the whole-body plethysmograph chamber. The outflow from the chamber was dried through a drierite column before passing through the $O_2$ analyzer where $O_2$ fraction in the outflow gas was continuously sampled (1000 Hz) and digitized via PowerLab (ADInstruments/Chart Software, version 7.3). The fractions of oxygen in the inflow ($FiO_2$) and outflow ($FeO_2$) gas were measured using a gas analyzer (model ML206, ADInstuments) that sampled, alternatively from the input and outflow gas ports. $O_2$ consumption ($VO_2$) was calculated based on the formula (*Depocas and Hart, 1957*): $VO_2$ = [FLo($FiO_2$ – $FeO_2$)]/1 – $FiO_2$, where FLo is the outlet flow rate; $FiO_2$ is the inflow $O_2$ fraction; $FeO_2$ is the outflow $O_2$ fraction. $VO_2$ was divided by body mass (in g) and the values reported under standard temperature and pressure, dry (STPD).

## Histology

The mice were deeply anesthetized with isoflurane (5% in 100% $O_2$) and heparin was injected intracardially (500 units) and perfused through the ascending aorta with 20 mL of phosphate-buffered saline (PBS 0.1 M) and with 50 mL of 4% paraformaldehyde (in PBS 0.1 M). The brains were kept overnight immersion in 4% paraformaldehyde and then in a 20% sucrose solution. Brain tissues were sectioned in a coronal plane at 30 μm with a sliding microtome and stored in cryoprotectant solution (20% glycerol plus 30% ethylene glycol in 50 mM phosphate buffer, pH 7.4) at –20°C until histological processing. All histochemical procedures were completed using free-floating sections.

For immunofluorescence, the following primary antibodies were used: (a) anti-*Phox2b* (rabbit anti-Phox2b 1:1000; a gift from JF Brunet, Ecole Normale Supèrieure, Paris, France); (b) anti-TH (mouse anti-TH, 1:1000; Millipore, Burlington, MA, USA); (c) anti-fos (rabbit anti-fos, 1:1000; Santa Cruz Biotechnology, Santa Cruz, CA, USA). All primary antibodies were diluted in PBS containing 2% normal donkey serum (Jackson ImmunoResearch Laboratories) and 0.3% Triton X-100 and were incubated overnight. Sections were subsequently rinsed in PBS and incubated for 2 hr in an appropriate secondary antibody (1:500). The sections were mounted in slides and covered with DPX (Sigma-Aldrich, Milwaukee, WI, USA).

## Mapping

A series of three 30 μm transverse sections through the brainstem were examined for each experiment using a Zeiss AxioImager A1 microscope (Carl Zeiss Microimaging, Thornwood, NY, USA). Images were taken with a Zeiss MRC camera (resolution 1388×1040 pixels). Only cell profiles that included a nucleus were counted and/or mapped bilaterally. Balance and contrast were adjusted to reflect true rendering as much as possible. No other 'photo retouching' was performed.

The total number of $Phox2b^+$, $Phox2b^+/TH^+$, and $Phox2b^+/TH^-$ cells in the parafacial/RTN region (between 5.99 and 6.75 mm caudal to Bregma level) was plotted as the mean ± SEM (8 sections/animal). We also analyzed $fos^+$ and $TH^-$ cells in the parafacial/RTN region. The neuroanatomical nomenclature employed during experimentation and in this manuscript was defined by the Mouse Brain Atlas from *Franklin and Paxinos, 2015*.

## Experimental protocols

### Experiment 1: Effect of $Phox2b^{\Delta 8}$ mutation in $Atoh1^{cre}$-expressing cells on breathing and chemoreflex activation during neonatal phase

Pups were placed in the plethysmography chambers (head-out or whole-body system) and acclimated 5 min prior to the experiment. To record breath parameters in the whole-body system, the flow was interrupted, and the chamber was closed for 1 min. We recorded a total of 3–5 min of ventilation in room air to determine the baseline. To induce chemoreflex challenge, pups were ventilated during 5 min in hypercapnia (7% $CO_2$, 21% $O_2$, balance $N_2$) or hypoxia (8% $O_2$, balance $N_2$) separated by a 10 min of recovery period (room air). In a separate experiment, we also measure $VO_2$ in neonates to investigate whether any change in body weight and baseline respiratory parameters might be related to changes in metabolic rate.

### Experiment 2: Effect of $Phox2b^{\Delta 8}$ mutation in $Atoh1^{cre}$-expressing cells on breathing and chemoreflex activation during adult phase

Adult mice were familiarized during 30 min in 3 consecutive days in the plethysmography chambers (whole-body system). At the day of the breathing recording, animals were acclimated 30–45 min prior to the experiment. After this acclimation, we recorded 10 min in room air breathing to determine the baseline. Animals were then exposed to hypercapnia or hypoxia during 10 min separated by a 20 min of recovery period in room air. In a separate experiment, we also measure $VO_2$ in adults to investigate whether any change in body weight and baseline respiratory parameters could be related to changes in metabolic rate.

## Experiment 3: Anatomical changes induced by $Phox2b^{\Delta8}$ mutation in the parafacial/ RTN region

To investigate whether $Phox2b^{\Delta8}$ mutation compromised *Phox2b* expression in the parafacial/RTN neurons, adult mice were anesthetized and perfused transcardially. Next, tissues were processed by immunohistochemistry to identify *Phox2b* expression and absence of TH (see details in Histology section).

## Experiment 4: Effect of hypercapnia on fos expression in the parafacial/RTN neurons induced by $Phox2b^{\Delta8}$ mutation

To investigate whether $Phox2b^{\Delta8}$ mutation compromised the activation of parafacial/RTN neurons by hypercapnia, we analyze fos expression in adult mice. Animals were habituated in the plethysmography chambers and ventilated in room air (0.5 L/min) during 3 consecutive days. At the day of experiment, mice were acclimated 1 hr prior to the hypercapnic challenge. Then, animals were exposed to hypercapnia (7% $CO_2$, 21% $O_2$, balance $N_2$) for 45 min. After exposure, mice were ventilated for additional 45 min in room air. Finally, animals were anesthetized and perfused transcardially as described above in Histology section. All experiments were conducted between 9:00 a.m. and 3:00 p.m.

## Statistical analysis

Results are presented as mean ± SEM. All statistics were performed using GraphPad Prism (version 9, GraphPad Software), with parametric tests used for normally distributed datasets. Details of specific tests are provided in the legend of each figure. The significance level was set as $p < 0.05$.

## Acknowledgements

This research work was supported by public funding from São Paulo Research Foundation (FAPESP) (Grants: 2019/01236-4 to ACT and 2015/23376-1 to TSM), and by funds from FAPESP fellowship (2017/12678-2 to TMS and 2019/20990-1 to PES), Conselho Nacional de Desenvolvimento Científico e Tecnológico (CNPq) grant (408647/2018-3 to ACT) and fellowships (302334/2019-0 to TSM and 302288/2019-8 to ACT) and NHLBI/NIH (Grant: R01HL132355 to CMC and JJO). This study was financed in part by the Coordenação de Aperfeiçoamento de Pessoal de Nível Superior-Brasil (CAPES) – Finance Code 001.

## Additional information

### Funding

| Funder | Grant reference number | Author |
|---|---|---|
| Fundação de Amparo à Pesquisa do Estado de São Paulo | 2009/01236-4 | Ana C Takakura |
| Fundação de Amparo à Pesquisa do Estado de São Paulo | 2015/23376-1 | Thiago S Moreira |
| NHLBI Division of Intramural Research | RO1HL132355 | José J Otero |
| Conselho Nacional de Desenvolvimento Científico e Tecnológico | 302334/2019-0 | Thiago S Moreira |
| Conselho Nacional de Desenvolvimento Científico e Tecnológico | 302288/2019-8 | Ana C Takakura |

The funders had no role in study design, data collection and interpretation, or the decision to submit the work for publication.

## Author contributions

Caroline B Ferreira, Conceptualization, Data curation, Software, Formal analysis, Validation, Investigation, Visualization, Methodology, Writing - original draft, Writing - review and editing; Talita M Silva, Conceptualization, Data curation, Formal analysis, Validation, Investigation, Visualization, Methodology, Writing - original draft, Writing - review and editing; Phelipe E Silva, Formal analysis, Validation, Investigation, Visualization, Methodology, Writing - review and editing; Claudio L Castro, Data curation, Formal analysis, Investigation, Methodology, Writing - review and editing; Catherine Czeisler, Conceptualization, Resources, Funding acquisition, Project administration, Writing - review and editing; José J Otero, Conceptualization, Resources, Data curation, Software, Formal analysis, Funding acquisition, Project administration, Writing - review and editing; Ana C Takakura, Thiago S Moreira, Conceptualization, Resources, Data curation, Software, Formal analysis, Supervision, Funding acquisition, Validation, Investigation, Visualization, Methodology, Writing - original draft, Project administration, Writing - review and editing

## Author ORCIDs

Thiago S Moreira (iD) http://orcid.org/0000-0002-9789-8296

## Ethics

This study was conducted in accordance with the University of Sao Paulo Institutional Animal Care and Use Committee guidelines (protocol number: 3618221019).

## Decision letter and Author response

Decision letter https://doi.org/10.7554/eLife.73130.sa1
Author response https://doi.org/10.7554/eLife.73130.sa2

---

# Additional files

## Supplementary files

• Transparent reporting form

## Data availability

All data generated or analyzed during this study are included in the manuscript and supporting file; Source Data files have been provided for Figures 1-6.

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
