## [Editor Report]

The present manuscript by Ferreira and colleagues is of potential interest to researchers working in the field of neural control of breathing and associated respiratory disorders. This study provides some novel insight into some genetic lesions that may underlie some developmental respiratory pathophysiologies.

---

## [Decision Letter]

**Decision letter after peer review:**

Thank you for submitting your article "*Phox2b* mutation mediated by *Atoh1* expression impaired respiratory rhythm and ventilatory responses to hypoxia and hypercapnia" for consideration by *eLife*. Your article has been reviewed by 3 peer reviewers, and the evaluation has been overseen by a Reviewing Editor and Martin Pollak as the Senior Editor. The following individuals involved in review of your submission have agreed to reveal their identity: Hiroshi Onimaru (Reviewer #1); Luis Rodrigo Hernandez-Miranda (Reviewer #3).

Essential revisions:

The present manuscript by Ferreira and colleagues is of potential interest to researchers working in the field of neural control of breathing and associated respiratory disorders. This study provides some novel insight into some genetic lesions that may underlie some developmental respiratory pathophysiologies. However, limitations in novelty, methodological approaches and data interpretations prevent acceptance of the paper in its present form.

To sum,

1 – Despite the fact that the authors propose a new genetic strategy to manipulate the development of the RTN in mice, there is a limited level of novelty when compared to previous already published works that have already clearly established the importance of Phox2b and Atoh1 regarding their role in specifying RTN constitutive neurons with their functions in central chemoception and breathing rhythm in general. You should strongly and deeply argue and discuss the differences between your present results and already known data addressing the role of Phox2b and Atoh1 in the RTN function and respiratory disorders.

2 – In addition, there are several methodological aspects and interpretation that deserve attention: the most important one being the practices for plethysmographic recordings in newborn and adult mice that must be performed in correct conditions in order to be able to draw strong and unambiguous conclusions; the identity of cFos-positive cells should be better described (are they indeed Phox2b+?). A fate map of GFP-expressing cells is required, including at different developmental stages. This is crucial as a precedent study has shown mislocation of RTN cells after genetic manipulations, this possibility is by the way not documented, examined or discussed. In addition, the status of other structures co-expressing Phox2b and Atoh1 must be described, as they might also affect breathing behavior.

3 – Illustrations must be ameliorated (Figure 1 is useless, Figure 4 requires traces before gas challenges and back to control, Fig8: the VII nucleus must be Phox2b positive) and additional data should be presented (map of GFP+ cells as stated above, activity of RTN cells during challenges).

*Reviewer #1 (Recommendations for the authors):*

Authors introduced new strategy of genetic manipulation in mice to reveal functional development of the RTN neurons that may relate to CCHS neuropathology. The methods and results are fairly clear. However, it might be rather difficult and complicated work to discuss the present findings because the results should be compared with those of various previous works including genetic manipulations of the RTN. I think that authors struggled to these issues, but I appreciate if they did more effort for this.

General comments

To understand strategy of this study, it would be required to know precise relationship between Phox2b and Atoh1 expression in the RTN during development: e.g. whether Phox2b expression requires preceding Atoh1 expression or not, whether Atoh1 is not required for Phox2b expression but does for correct localization of Phox2b expressing cells, etc. Although some information was mentioned, please add a brief further explanation for this issue in the Introduction to facilitate understanding of their strategy for general readers (see also comments for Discussion section).

In addition, it would be important to correctly understand differences between genetic removal and induction of (conditional) mutation of Phox2b (or Atoh1). It seems to be rather complicated for general readers to understand these issues and what are new findings of the present study, because various genetic manipulations to investigate the RTN function have been performed.

Specific comments

Abstract

The last sentence, "…of the RTN neurons and are essential for the activation of breathing under hypoxic and hypercapnia condition…": What are essential for the activation? (Same as the last sentence in the Introduction).

Methods

Authors described "All experiments, were conducted between 9:00 A.M. and 3:00 P.M."

How do you think about the sleep-wake cycle condition of mice during measurements? This might be important in considering relation to CCHS.

Results

Although the number of specimens for statistical analysis is described in Figure 2 legend, it is not clear whether the same number could be applied in other figures. For instance, the number in Figure 3 seems to be different from those in Figure 2 (e.g. in neonate control and adult control). Moreover, the specimen number for body weight analysis was not described (Page 10, para 3). Please check the number of all experiments. It would be better to describe them in each figure or in the text.

Section 6: The fos expression was counted in RTN neurons lacking TH immunoreactivity. Is it correct that these cells are Phox2b-positive?

Section 7: The results showed that number of Phox2b expressing cells in mutant mice was about half of that in control mice. Authors should discuss why these results were obtained, because it was previously reported that almost 100% of Atoh1 expressing cells were also Phox2-positive (Dubreuil et al. 2009). Did the present results indicate that 50 % of Phox2b expressing cells in the mutant RTN were Atoh1 independent? Or some of Phox2b△8 mutated cells are remaining?

Section 7: Did authors confirmed that Phox2b-positive and TH-positive cells in the medulla and pons were not affected by this mutation?

Discussion

If authors could summarize possible relationships between Atoh1 and Phox2b (and maybe other related transcription factors) involved in the RTN development, considering previous and present results, it would be very helpful for understanding the present situation and future problem of this field. I strongly recommend that authors give a figure for this purpose.

*Reviewer #2 (Recommendations for the authors):*

The manner in which the plethysmography for both the neonates and adults is not technically sound. There are several problems with their approaches. For neonates, whole body plethysmography is not an appropriate method to measure tidal volume. As the authors themselves note in their methods section, whole body plethysmography is used "to detected pressure oscillations as a result of changes in temperature promoted by ventilation when chamber was completely closed" [sic]. In neonate plethysmography, P1-3 pups typically equilibrate with their environmental chamber temp. Thus, the waveform is not a derivative of heating and cooling that can be related to tidal volume by applying Bartlett and Tenney corrections but rather a function of airway resistance as the compression in the chamber equilibrates with the rarefaction in the lungs upon inspiration (PMID: 25017785). This phenomenon is also a component of adult respiratory measurements, but less so with a large enough temperature differential (30-33 chamber temp) making the barometric component the predominant feature. Facemask pneumotachography or headout plethysmography can give a more accurate and consistent estimate of tidal volume in neonates.

In adult plethysmography, several necessary measurements were not taken or presented. Body temperatures is not reported, nor are VO2 or VCO2. In plethysmography, especially in chemosensory studies, these are critical measurements that should be taken concurrently with breathing measurements and reported singularly and as VE/VO2. If any metabolic or temperature differences exist compared to control groups, this will have significant impacts on breathing outcomes. Changes in metabolism can drive persistent states of alkalosis and acidosis that would impact responses to chemosensory challenges (though such states may be mitigated by renal compensation). Metabolism is state dependent and plethysmography, even with habituation, is still stressful (PMID: 31178741). As noted by authors and further elaborated by Frappel and Mortola (PMID: 1621857, VO2 and temperature will drop in response to hypoxia. Additionally, poikilocapnic hypoxic hyperventilation (vs isocapnic)) causes a drop in pCO2 that reduces drive to breath (PMID: 23690557). Lastly, the chamber temperature for the adult studies should be held at thermo-neutrality rather than room temperature (30-33C). The cold challenge to mice that is room temperature has confounding effects on drive to breath and metabolism. By measuring adults in thermo-neutral conditions, potential metabolic effects are minimized. Isocapnic hypoxia may also be considered.

Given that there is a reported body weight difference and the authors are using a cold chamber for adults, there very well may be an unappreciated difference in metabolism or in metabolic changes due to changes in state, response to hypoxia, or the cold challenge that impacts the reported outcomes. Metabolic differences may also arise (leading to weight loss) through other Phox2B – Atoh1 overlapping populations perturbed in this model not considered in the manuscript.

The observed phenotypes cannot be exclusively assigned to the RTN. A full assessment of Atoh1 and Phox2b overlap using cumulative fatemapping afforded by the Atoh1_Cre; Phox2bΔ8 model should be reported as other areas of overlap could either impact breathing directly or indirectly through metabolism and stress responses (PMID 8184995). The role of the previously identified Atoh1; Phox2B para and intra – trigeminal neurons should be accounted for in the phenotypes and/or the Atoh1; Phox2B RTN neurons tested in isolation.

*Reviewer #3 (Recommendations for the authors):*

Ferreira and colleagues provide a novel mouse model (Atoh1Cre,Phox2bdelta8) for the study of the central respiratory chemoreceptor circuit and, therefore, of interest for the respiratory physiology community. Nonetheless, in its present form, this work still lacks more physiological, developmental, and anatomical characterizations to place this study in a broader context and gain new insights into the physiology of respiratory chemoreflexes.

I hope the authors find my below comments of use to enrich their work.

1. The major caveat of this work is that it does not significantly differ from previously published reports using very similar approaches (including a Atoh1Cre,Phox2bflox/flox strategy in Ruffault et al., 2015, *eLife* DOI: 10.7554/*eLife*.07051).

2. For today's standards, the display of the alleles (genetic strategy) used in this study cannot occupy a full main figure (Figure 1).

3. The plethysmograph traces presented in Figures 2, 4, 5 and 6 should be accompanied by a period of pre-Gas exposure and post-Gas exposure.

4. Figure 7 and 8 could be combined, and more representative photographs should be presented. The assignment of the facial motor nucleus seems to be arbitrary, as it lacks Phox2b immunoreactive cells. Facial motor neurons do express Phox2b in addition to ChAT in the adult life of mice.

5. The general outline of the manuscript is nothing I have seen before in *eLife*, that is numbered points for Materials and methods and the result section, although I admit that it helped a lot with the reading.

6. The authors show a reduction (about half) in the number of Phox2b+/TH- cells in adult Atoh1Cre,Phox2bdelta8 mice, and assume that this is indicative of a reduction in the number of retrotrapezoid neurons. This is not necessarily true. I would recommend that the authors present first an anatomical/developmental characterization of retrotrapezoid neurons in Atoh1Cre,Phox2bdelta8 at embryonic and neonatal states (E12.5, E16.5 and P0). At this stages, retrotrapezoid neurons have a well-established molecular signature: Phox2b, Atoh1 and Lbx1 expression. Whereas there are not great commercial Lbx1 and Atoh1 antibodies, the authors could consider combining in situ hybridization for Lbx1 and/or Atoh1 with Phox2b immunoreactivity, the Phox2b antibody that the authors used in this study is great, and compatible with in situ hybridization (my own experience). Other studies by the groups of Huda Zoghbi and Jean-Francois Brunet have shown that interfering with Phox2b and Atoh1 expression in retrotrapezoid neurons results in the incorrect location of this cells dorsally to the facial motor nucleus, is this phenotype also present in Atoh1Cre,Phox2bdelta8 mice?

7. The authors show a reduction of Fos+/TH- cells in adult Atoh1Cre,Phox2bdelta8 mice that were exposed to high levels of CO2 in air. From this result the authors conclude that the number of Fos-activated retrotrapezoid neurons is decreased. Again, this is not necessarily true. To better define this, it is necessary to demonstrate that Fos is not express in Phox2b+/TH- retrotrapezoid neurons. Whereas the Fos and Phox2b antibodies used in this study are both generated in rabbits, the authors could make use of the eGFP expression present in the Phox2bdelta8 allele (Figure 1), therefore, it is necessary to combine the immunoreactivity for eGFP (Phox2b), Fos and TH. This is central for this study, as if indeed less retrotrapezoid neurons are activated by CO2 in adult Atoh1Cre,Phox2bdelta8 mice, it is astonishing that these mice can have a full response to hypercapnia. This is intriguing because other mouse models, in which the number of retrotrapezoid neurons are reduced in greater numbers, do not show a full response to CO2 in the adult life, for instance in: P2b::CreBAC1;Atoh1lox/lox (Ruffault et al., 2015), Egr2cre;P2b27Alacki (Ramanantsoa et al., 2011, DOI: 10.1523/JNEUROSCI.1721-11.2011), Atoh1Phox2bCKO mice (Huang et al., 2017, DOI: 10.1016/j.neuron.2012.06.027) and Egr2cre;Lbx1FS (Hernandez-Miranda et al., 2018, DOI: 10.1073/pnas.1813520115).

8. The authors do not address if retrotrapezoid neurons/parafacial cells are rhythmically active and responsive for pH changes in the embryonic/neonatal life of Atoh1Cre,Phox2bdelta8 animals. The fact that these mice can fully respond to hypercapnia in the adult life but not in the neonatal stages might imply that retrotrapezoid neurons are present but somehow silenced in Atoh1Cre,Phox2bdelta8 neonates. Therefore the anatomical/developmental characterization of retrotrapezoid neurons (as suggested above) should be complemented with in vitro calcium imaging in Atoh1Cre,Phox2bdelta8 embryos or neonates. This could explain why mice that completely lack retrotrapezoid neurons (Egr2cre;Lbx1FS) do not fully display the hypercapnic reflex, whereas Atoh1Cre,Phox2bdelta8 mice do.

---

## [Author Response]

Essential revisions:The present manuscript by Ferreira and colleagues is of potential interest to researchers working in the field of neural control of breathing and associated respiratory disorders. This study provides some novel insight into some genetic lesions that may underlie some developmental respiratory pathophysiologies. However, limitations in novelty, methodological approaches and data interpretations prevent acceptance of the paper in its present form.

We thank the reviewers and the editors for the constructive comments on our manuscript. We have carefully addressed most of the major comments. We modified the manuscript accordingly. We believe that the revised version of our manuscript has been greatly improved and we hope that it will be suitable for publication on the *eLife*.

To sum,1 – Despite the fact that the authors propose a new genetic strategy to manipulate the development of the RTN in mice, there is a limited level of novelty when compared to previous already published works that have already clearly established the importance of Phox2b and Atoh1 regarding their role in specifying RTN constitutive neurons with their functions in central chemoception and breathing rhythm in general. You should strongly and deeply argue and discuss the differences between your present results and already known data addressing the role of Phox2b and Atoh1 in the RTN function and respiratory disorders.

We would like to thank the reviewers and editor for the important concern and the opportunity to better discuss the differences between our study and former studies that have already described the role of Phox2b mutation or genetic deletion on chemoreception and breathing regulation. We implemented several modifications in the manuscript to address this question. Briefly, the main novelty of our study was to introduce a humanized NPARM Phox2b^Δ8^ in Atoh1^cre^-expressing cells of rodents. Atoh1 is expressed during development in proliferating cells in the rhombic lip and in postmitotic neurons. In this independent site, postmitotic neurons are the only region that co-express Phox2b and Atoh1 surround the paramotor neurons that involves facial motor nucleus (peri VII thus, RTN/parafacial neurons) and trigeminal motor nucleus (periV also known as intertrigeminal region). The present study highlighted some differences compared to deletion of a transcription factor (Phox2b or Atoh1) or introduction of PARM CCHS mutation revealed by previous studies. Surprisingly, (1) Phox2bΔ8 in Atoh1expressing cells did not induce important functional change of baseline respiration in neonates. In contrast, it increased the number of apneas and respiratory irregularity in adult mice. (2) Ventilatory responses to hypoxia and hypercapnia are compromised in neonates. Interestingly, same lack of hypercapnic response have been demonstrated after introducing PARM mutation specific to RTN (DOI:10.1523/JNEUROSCI.1721-11.2011) or by deleting Phox2b from same Phox2b/Atoh1 expressing neurons (periV and peri VII) (DOI: 10.7554/*eLife*.07051.001). However, the compromised ventilatory response to hypoxia showed by us contrast with the study that introduced PARM mutation in the RTN (DOI:10.1523/JNEUROSCI.1721-11.2011). Furthermore, (3) in adults, the ventilatory response to hypoxia still partially impaired. But they recovered hypercapnic ventilatory response. Despite hypercapnic responses is recovered, the number of Phox2b+/TH- neurons in the RTN/parafacial region reduced by approximately 50%. Additionally, the number of fos+/TH- expressing neurons in the RTN/parafacial region induced by hypercapnia drastically reduced in adults. Interestingly, formers studies that applied different strategies to manipulate Phox2b RTN/parafacial neurons, CO_2_ response was only partially recovered in the adult life (DOI: 10.7554/*eLife*.07051; DOI:10.1523/JNEUROSCI.1721-11.2011; http://dx.doi.org/10.1016/j.neuron.2012.06.027; https://doi.org/10.1073/pnas.1813520115). Although, an extensive depletion of RTN/parafacial neurons occurred at embryonic ages. There is no information whether those neurons still depleted during adulthood and whether they are functional. Thus, it complicates further comparison with our finding. Together these results highlighted important differences that certainly imply different mechanisms between our strategy and previous studies. The mechanism by which NPARM Phox2bΔ8 mutation impact ventilatory responses induced by hypoxia and hypercapnia will be focus of future studies in the laboratory.

2 – In addition, there are several methodological aspects and interpretation that deserve attention: the most important one being the practices for plethysmographic recordings in newborn and adult mice that must be performed in correct conditions in order to be able to draw strong and unambiguous conclusions; the identity of cFos-positive cells should be better described (are they indeed Phox2b+?). A fate map of GFP-expressing cells is required, including at different developmental stages. This is crucial as a precedent study has shown mislocation of RTN cells after genetic manipulations, this possibility is by the way not documented, examined or discussed. In addition, the status of other structures co-expressing Phox2b and Atoh1 must be described, as they might also affect breathing behavior.

Thank you for several comments on methodology. We acknowledge the limitations of the barometric plethysmography for the precise measurement of tidal volume. For that reason, breath volumes were normalized to calibrations made during each recording. The amplitude ratio of breaths compared to calibrations was used to determine a value in microliter for each breath. In addition, modifications induced by hypoxia/hypercapnia stimulus was showed as percentage change and not absolute values. To account for differences in body size, breath volumes were also normalized to body weight (μl/g). As a result, we do not expect these limitations to have a significant impact on the interpretation of our data. In addition, to further investigate whether the plethysmography method used would change the former tidal volume results, specialty during baseline conditions where the absolute values were described, we performed head-out plethysmograph system in a subset of neonate (control and mutant). We found that neonate mutants mice presented a slightly increase in tidal volume. However, it did not reach statistic differences, presumably due to the small number of the sample (N = 4/group). In any case, we added the results in the result section and mentioned in the text the importance to use more accurate methods since it has not been used by majority of the studies, including those that we referenced in the present study (Mortola, 1984, Durand et al., 2004; Mortola and Frappell, 2013; Patrone et al., 2018).

In relation to the fos expression, we better described our protocol and clarify the methodology used in the text (methods, results and Discussion section). We used the absence of coexpression with tyrosine hydroxylase, as it has been extensively described to identify RTN activated neurons located ventrally to facial motor nucleus (Stornetta et al., 2006. Barna et al., 2012; 2014; Kumar et al., 2015). Unfortunately, the fos and Phox2b antibodies that we have in the laboratory are made in the same species (i.e, rabbit) and could not be ran together. Furthermore, we investigated RTN/parafacial neurons by analyzing the presence of Phox2b and absence of TH expression. Phox2b+/TH- neurons are well known to be chemosensitive RTN neurons (Stornetta et al., 2006; Onimaru et al., 2008). We found that the number of Phox2b+/TH- neurons greatly reduced in adult mutant mice. Importantly, the neurons were located ventral and laterally to facial motor nucleus. In agreement with previous study that deleted Phox2b from same neuronal region (DOI: 10.7554/*eLife*.07051.001). More specifically, from Phox2b/Atoh1 post mitotic neurons in the peri V and periVII region. Interestingly, the misslocation only occurred when manipulating Atoh1 neurons (DOI: 10.7554/*eLife*.07051.001; DOI:https://doi.org/10.1016/j.neuron.2012.06.027 ).

Regarding the fate map of GFP-expressing cells, in concordance with our prior data (DOI 10.1007/s00401-015-1441-0; doi:10.1111/bpa.12877), GFP expression is only detected in embryonic Phox2b expressing cells. The goal of the present study was to analyze respiratory and anatomical effect after birth and not during embryonic phase. For that reason, a fate map showing GFP-expressing cells during different developmental stages was not performed.

3 – Illustrations must be ameliorated (Figure 1 is useless, Figure 4 requires traces before gas challenges and back to control, Fig8: the VII nucleus must be Phox2b positive) and additional data should be presented (map of GFP+ cells as stated above, activity of RTN cells during challenges).

Figure 1 was removed. We edited and added traces showing pre and pos stimulus. Regarding the Phox2b positive cells in the VII nucleus, we do not have a final answer why there is no Phox2b-immunoreactive in the facial motor nucleus. However, previous experiments from our lab (PMID: 22015927, 24286756, 24363384, 27633663) and from different labs (PMID: 17021186, 17255166, 18440993, 26068853, 29066557) did not find Phox2b-expressing neurons in the facial motor nucleus of postnatal or adult rodents.

Reviewer #1 (Recommendations for the authors):Authors introduced new strategy of genetic manipulation in mice to reveal functional development of the RTN neurons that may relate to CCHS neuropathology. The methods and results are fairly clear. However, it might be rather difficult and complicated work to discuss the present findings because the results should be compared with those of various previous works including genetic manipulations of the RTN. I think that authors struggled to these issues, but I appreciate if they did more effort for this.

We agree with the reviewer. In the present version of the manuscript, we better discussed our data in comparison with former studies. We rewrote the manuscript and most of the information was added in the introduction and Discussion section. Briefly, the main novelty of our study was to introduce a humanized NPARM Phox2b^Δ8^ in Atoh1^cre^-expressing cells of rodents. Atoh1 is expressed during development in proliferating cells in the rhombic lip and in postmitotic neurons. In this independent site, postmitotic neurons are the only region that co-express Phox2b and Atoh1 surround the paramotor neurons that involves facial motor nucleus (peri VII thus, RTN/parafacial neurons) and trigeminal motor nucleus (periV also known as intertrigeminal region). The present study highlighted some differences compared to deletion of a transcription factor (Phox2b or Atoh1) or introduction of PARM CCHS mutation revealed by previous studies.

Surprisingly, (1) Phox2bΔ8 in Atoh1-expressing cells did not induce important functional change of baseline respiration in neonates. In contrast, it increased the number of apneas and respiratory irregularity in adult mice. (2) Ventilatory responses to hypoxia and hypercapnia are compromised in neonates. Interestingly, same lack of hypercapnic response have been demonstrated after introducing PARM mutation specific to RTN (DOI:10.1523/JNEUROSCI.1721-11.2011) or by deleting Phox2b from same Phox2b/Atoh1 expressing neurons (periV and peri VII) (DOI: 10.7554/*eLife*.07051.001). However, the compromised ventilatory response to hypoxia showed by us contrast with the study that introduced PARM mutation in the RTN (DOI:10.1523/JNEUROSCI.1721-11.2011). Furthermore, (3) in adults, the ventilatory response to hypoxia still partially impaired. But they recovered hypercapnic ventilatory response. Despite hypercapnic responses is recovered, the number of Phox2b+/TH- neurons in the RTN/parafacial region reduced by approximately 50%. Additionally, the number of fos+/TH- expressing neurons in the RTN/parafacial region induced by hypercapnia drastically reduced in adults. Interestingly, formers studies that applied different strategies to manipulate Phox2b RTN/parafacial neurons, CO_2_ response was only partially recovered in the adult life (DOI: 10.7554/*eLife*.07051; DOI:10.1523/JNEUROSCI.1721-11.2011; http://dx.doi.org/10.1016/j.neuron.2012.06.027; https://doi.org/10.1073/pnas.1813520115). Although, an extensive depletion of RTN/parafacial neurons occurred at embryonic ages. There is no information whether those neurons still depleted during adulthood and whether they are functional. Thus, it complicates further comparison with our finding. Together these results highlighted important differences that certainly imply different mechanisms between our strategy and previous studies. The mechanism by which NPARM Phox2bΔ8 mutation impact ventilatory responses induced by hypoxia and hypercapnia will be focus of future studies in the laboratory.

General commentsTo understand strategy of this study, it would be required to know precise relationship between Phox2b and Atoh1 expression in the RTN during development: e.g. whether Phox2b expression requires preceding Atoh1 expression or not, whether Atoh1 is not required for Phox2b expression but does for correct localization of Phox2b expressing cells, etc. Although some information was mentioned, please add a brief further explanation for this issue in the Introduction to facilitate understanding of their strategy for general readers (see also comments for Discussion section).

In the present study, we introduced the humanized NPARM Phox2b^Δ8^ in Atoh1expressing cells of rodents. Atoh1 in the brainstem is expressed during development in proliferating cells in the rhombic lip (at early stages) and in postmitotic neurons (during late stage) independently. The postmitotic neurons are the only region that co-express Phox2b and Atoh1 surround the paramotor neurons that involves facial motor nucleus (peri VII thus, RTN/parafacial neurons) and trigeminal motor nucleus (periV also known as intertrigeminal region) (DOI: 10.7554/*eLife*.07051.001). Thus, NPARM Phox2b^Δ8^ was introduced in the periVII and periV region. Our goal with this strategy was to investigate the effect of the NPARM Phox2b^Δ8^ in regions that are well known to be involved with respiratory control and central chemosensitivity. The idea was to induce late activation of NPARM Phox2b^Δ8^ and, then analyze the effect on respiratory function and neuroanatomy of RTN after birth (neonatal and adulthood). The results in the present study showing immunoreactivity to Phox2b, and absence of tyrosine hydroxylase strongly indicate the phenotype of RTN/parafacial chemosensitive neurons. Furthermore, we showed that when the mutation is activated in the postmitotic Phox2b/Atoh1 neurons, it greatly impacts RTN/parafacial neurons (reduction in approximately 50% compared to controls). Deletion of Phox2b using similar strategy (from same periV and periVII region) reduced Phox2b+/Atoh1+ expression by 60% during embryonic age 18.5.

We also added more information in the Introduction and Discussion to be clear for general readers.

In addition, it would be important to correctly understand differences between genetic removal and induction of (conditional) mutation of Phox2b (or Atoh1). It seems to be rather complicated for general readers to understand these issues and what are new findings of the present study, because various genetic manipulations to investigate the RTN function have been performed.

We apologize for the confusion and have better described the genetic approach used. The present study used a conditional “humanized” Phox2b NPARM mutation inspired by a proband previously reported by Dr. Otero´s group. This locus includes a human exon 3 mutation with an 8-nucleotide deletion. It is not induced by Atoh1. Rather, cells that express Atoh1 undergo the cre-mediated recombination at the engineered Phox2b locus and will express the NPARM phox2b only when the gene is normally expressed. Phox2b shows expression early in mammalian development, and therefore this construct provides for the study of NPARM Phox2b effects in respiratory nuclei after initial embryonic Phox2b specification.

Specific commentsAbstractThe last sentence, "…of the RTN neurons and are essential for the activation of breathing under hypoxic and hypercapnia condition…": What are essential for the activation? (Same as the last sentence in the Introduction).

We modified the entire Abstract section.

MethodsAuthors described "All experiments, were conducted between 9:00 A.M. and 3:00 P.M."How do you think about the sleep-wake cycle condition of mice during measurements? This might be important in considering relation to CCHS.

Despite we have done all recordings during light cycle where the percentage of sleep is higher, unfortunately, we did not record it concomitant with respiration. So, we cannot guarantee if animals were sleeping or not during the breathing recordings. Some preliminary data from our lab (not shown in this manuscript) found that percentage of sleep-awake cycle was similar between control and mutated mice. We are planning to better evaluate in a future study a comparison of respiratory control between wake and sleep state in both control and mutant mice (Phox2b^Δ8^ mutation in Atoh1^cre^).

ResultsAlthough the number of specimens for statistical analysis is described in Figure 2 legend, it is not clear whether the same number could be applied in other figures. For instance, the number in Figure 3 seems to be different from those in Figure 2 (e.g. in neonate control and adult control). Moreover, the specimen number for body weight analysis was not described (Page 10, para 3). Please check the number of all experiments. It would be better to describe them in each figure or in the text.

We acknowledged the observation. In the present version, we checked all the numbers and statistics. We apologized for the mistake. In addition, we found that the mean of inter-breath intervals was done incorrectly, i.e., not converting to absolute values. So, we also correct it in the present version of the manuscript, figure and excel sheet. To clarify, none of the results or statistics had been changed after correcting those mistakes.

Section 6: The fos expression was counted in RTN neurons lacking TH immunoreactivity. Is it correct that these cells are Phox2b-positive?

We counted the number of fos+ neurons in the RTN/parafacial region that lack TH immunoreactivity. It is likely that these cells also express Phox2b because in this region we have the bulk of Phox2b-expressing neurons. We did not perform Phox2b and fos because our antibodies are made in rabbits. We avoid the TH cells, which also express Phox2b but it is not the chemoreceptors RTN neurons.

Section 7: The results showed that number of Phox2b expressing cells in mutant mice was about half of that in control mice. Authors should discuss why these results were obtained, because it was previously reported that almost 100% of Atoh1 expressing cells were also Phox2-positive (Dubreuil et al. 2009). Did the present results indicate that 50 % of Phox2b expressing cells in the mutant RTN were Atoh1 independent? Or some of Phox2b△8 mutated cells are remaining?

In the present version, we did not analyze the percentage of Atoh1 cells that colocalize with Phox2b in the RTN. Previous studies that used in situ hybridization showed that approximately all Phox2b neurons of the RTN also express Atoh1 (PMID: 22958821, 25866925). We used the cre line as a tool to introduce the conditional Phox2b mutation during Atoh1 expression. Thus, cells that express Atoh1 undergo the cre-mediated recombination at the engineered Phox2b locus and will express the NPARM phox2b only when the gene is normally expressed. Phox2b shows expression early in mammalian development, and therefore this construct provides for the study of NPARM Phox2b effects in respiratory nuclei after initial embryonic Phox2b specification.

Section 7: Did authors confirmed that Phox2b-positive and TH-positive cells in the medulla and pons were not affected by this mutation?

Yes, we ran a new series of experiments and noticed that Phox2b+ neurons in the pons as well as the number of TH cells in the A1, A2, A6, and C1 were not affected by the mutation.

DiscussionIf authors could summarize possible relationships between Atoh1 and Phox2b (and maybe other related transcription factors) involved in the RTN development, considering previous and present results, it would be very helpful for understanding the present situation and future problem of this field. I strongly recommend that authors give a figure for this purpose.

Thank you for the suggestion. In the present version of the manuscript, we made a summary figure illustrating what we have in the literature and what our work added to the field of breathing control.

Reviewer #2 (Recommendations for the authors):The manner in which the plethysmography for both the neonates and adults is not technically sound. There are several problems with their approaches. For neonates, whole body plethysmography is not an appropriate method to measure tidal volume. As the authors themselves note in their methods section, whole body plethysmography is used "to detected pressure oscillations as a result of changes in temperature promoted by ventilation when chamber was completely closed" [sic]. In neonate plethysmography, P1-3 pups typically equilibrate with their environmental chamber temp. Thus, the waveform is not a derivative of heating and cooling that can be related to tidal volume by applying Bartlett and Tenney corrections but rather a function of airway resistance as the compression in the chamber equilibrates with the rarefaction in the lungs upon inspiration (PMID: 25017785). This phenomenon is also a component of adult respiratory measurements, but less so with a large enough temperature differential (30-33 chamber temp) making the barometric component the predominant feature. Facemask pneumotachography or headout plethysmography can give a more accurate and consistent estimate of tidal volume in neonates.

We acknowledge the limitations of the barometric plethysmography for precise measurement of tidal volume in neonates used in the present study. For that reason, in a subset of control and mutant neonate mice we analyze respiration using head-out plethysmograph. Despite the small sample size, we found that tidal volume was slightly higher in the mutant compared to controls littermate. We added more information in the material and methods ((page 5; 2) Ventilation measurements), results (page 11, para 1^st^), and Discussion section (page 16, para 2^nd^). Additionally, in the experiments using whole body plethysmograph, breath volumes were normalized to calibrations made during each recording. The amplitude ratio of breaths compared to calibrations was used to determine a value in microliter for each breath. More importantly, changes induced by hypoxia/hypercapnia stimulus were showed as percentage change and not absolute values. To account for differences in body size, breath volumes were also normalized to body weight (μl/g). As a result, we do not expect that these limitations significantly impact the interpretation of our data.

In adult plethysmography, several necessary measurements were not taken or presented. Body temperatures is not reported, nor are VO2 or VCO2. In plethysmography, especially in chemosensory studies, these are critical measurements that should be taken concurrently with breathing measurements and reported singularly and as VE/VO2. If any metabolic or temperature differences exist compared to control groups, this will have significant impacts on breathing outcomes. Changes in metabolism can drive persistent states of alkalosis and acidosis that would impact responses to chemosensory challenges (though such states may be mitigated by renal compensation). Metabolism is state dependent and plethysmography, even with habituation, is still stressful (PMID: 31178741). As noted by authors and further elaborated by Frappel and Mortola (PMID: 1621857, VO2) and temperature will drop in response to hypoxia. Additionally, poikilocapnic hypoxic hyperventilation (vs isocapnic) causes a drop in pCO2 that reduces drive to breath (PMID: 23690557). Lastly, the chamber temperature for the adult studies should be held at thermo-neutrality rather than room temperature (30-33C). The cold challenge to mice that is room temperature has confounding effects on drive to breath and metabolism. By measuring adults in thermo-neutral conditions, potential metabolic effects are minimized. Isocapnic hypoxia may also be considered.

Thank you for pointing out important issues in our previous manuscript. We agree with the reviewer. In fact, some information and misconception were in the previous version. Now, we added the correct way in which the respiratory parameters were measured in both neonate and adult mice (Material and methods section; page 5 – ventilation measurements). We also measured VO2 and VE/VO2 in control and mutant mice during neonatal and adult phase. Despite the slightly reduction in body weight in the mutants during adulthood, oxygen consumption was not different between groups. Those results indicate that NPARM Phox2b^Δ8^ in Atoh1^cre^-expressing cells did not impact metabolism.

Given that there is a reported body weight difference and the authors are using a cold chamber for adults, there very well may be an unappreciated difference in metabolism or in metabolic changes due to changes in state, response to hypoxia, or the cold challenge that impacts the reported outcomes. Metabolic differences may also arise (leading to weight loss) through other Phox2B – Atoh1 overlapping populations perturbed in this model not considered in the manuscript.

As mentioned above some information and misconception were in the previous version. We performed additional experiments to analyze oxygen consumption. We found that mutation did not change VO2 and VE/VO2 in neonates nor in adults compare to controls. Thus, the reduction in body weight in adult mutant cannot be assigned to changes in the metabolism.

Unfortunately, most of the studies that genetically deleted Phox2b/Atoh1 (PMID: 25866925; PMID: 22958821; PMID: 29972353) did not report body weight. However, when PARM Phox2b mutation was induced specific to RTN neurons (PMID: 21900566), i.e., without directly manipulating periV region, mutated neonate (P9) had lower body weight when compared to controls. Nevertheless, oxygen consumption was not evaluated by the former study.

The observed phenotypes cannot be exclusively assigned to the RTN. A full assessment of Atoh1 and Phox2b overlap using cumulative fatemapping afforded by the Atoh1_Cre; Phox2bΔ8 model should be reported as other areas of overlap could either impact breathing directly or indirectly through metabolism and stress responses (PMID 8184995). The role of the previously identified Atoh1; Phox2B para and intra – trigeminal neurons should be accounted for in the phenotypes and/or the Atoh1; Phox2B RTN neurons tested in isolation.

We would like to thank the reviewer for the excellent point raised. Regarding the stress response, the study cited (PMID 8184995) demonstrated that children with CCHS had lower levels of anxiety compared to all other children, including those with asthma. However, there is no information about CCHS form (PARM or NPARM) included in the group. It will be an interesting topic to investigate in the future.

In relation to the cumulative fate mapping, we had several problems with our Phox2b^Δ8^; Atoh1^cre^ colony which prevented us to continue the experiments to show differences in the expression of Phox2b within the para or intraregional regions. We are trying to recover the colony and future experiments will be carried out to account for the phenotypes asked by the reviewer.

Reviewer #3 (Recommendations for the authors):Ferreira and colleagues provide a novel mouse model (Atoh1Cre,Phox2bdelta8) for the study of the central respiratory chemoreceptor circuit and, therefore, of interest for the respiratory physiology community. Nonetheless, in its present form, this work still lacks more physiological, developmental, and anatomical characterizations to place this study in a broader context and gain new insights into the physiology of respiratory chemoreflexes.I hope the authors find my below comments of use to enrich their work.

In the new version of the manuscript, we have made significant changes as suggested by the reviewer. For example, we reorganize the anatomical data including new analysis and performed new functional experiments to strengthen our work. We are very enthusiastic about our reviewed version, and we believe it will open new questions that need to be addressed in future studies

1. The major caveat of this work is that it does not significantly differ from previously published reports using very similar approaches (including a Atoh1Cre,Phox2bflox/flox strategy in Ruffault et al., 2015, eLife DOI: 10.7554/eLife.07051).

We would like to thank the reviewer for the important concerns and the opportunity to better discuss the differences between our study and former studies that have already described the role of Phox2b mutation or genetic deletion on chemoreception and breathing regulation. We implemented several modifications in the manuscript to address this question. We rewrote the manuscript and most of the information was added in the introduction and Discussion section. Briefly, the main novelty of our study was to introduce a humanized NPARM Phox2b^Δ8^ in Atoh1^cre^expressing cells of rodents. Atoh1 is expressed during development in proliferating cells in the rhombic lip and in postmitotic neurons. In this independent site, postmitotic neurons are the only region that co-express Phox2b and Atoh1 surround the paramotor neurons that involves facial motor nucleus (peri VII thus, RTN/parafacial neurons) and trigeminal motor nucleus (periV also known as intertrigeminal region). The present study highlighted some differences compared to deletion of a transcription factor (Phox2b or Atoh1) or introduction of PARM CCHS mutation revealed by previous studies. Surprisingly, (1) Phox2b^Δ8^ in Atoh1-expressing cells did not induce important functional change of baseline respiration in neonates. In contrast, it increased the number of apneas and respiratory irregularity in adult mice. (2) Ventilatory responses to hypoxia and hypercapnia are compromised in neonates. Interestingly, similar lack of hypercapnic response have been demonstrated after introducing PARM mutation specific to RTN (DOI:10.1523/JNEUROSCI.1721-11.2011) or by deleting Phox2b from same Phox2b/Atoh1 expressing neurons (periV and peri VII) (DOI: 10.7554/*eLife*.07051.001). However, the compromised ventilatory response to hypoxia showed by us contrast with the study that introduced PARM mutation in the RTN (DOI:10.1523/JNEUROSCI.1721-11.2011). Furthermore, (3) in adults, the ventilatory response to hypoxia still partially impaired. But they recovered hypercapnic ventilatory response. Despite hypercapnic responses is recovered, the number of Phox2b+/TH- neurons in the RTN/parafacial region reduced by approximately 50%. Additionally, the number of fos+/TH- expressing neurons in the RTN/parafacial region induced by hypercapnia drastically reduced in adults. Interestingly, formers studies that applied different strategies to manipulate Phox2b RTN/parafacial neurons, CO_2_ response was only partially recovered in the adult life (DOI: 10.7554/*eLife*.07051; DOI:10.1523/JNEUROSCI.1721-11.2011; http://dx.doi.org/10.1016/j.neuron.2012.06.027; https://doi.org/10.1073/pnas.1813520115). Although, an extensive depletion of RTN/parafacial neurons occurred at embryonic ages. There is no information whether those neurons still depleted during adulthood and whether they are functional. Thus, it complicates further comparison with our finding. Together these results highlighted important differences that certainly imply different mechanisms between our strategy and previous studies. The mechanism by which NPARM Phox2bΔ8 mutation impact ventilatory responses induced by hypoxia and hypercapnia will be focus of future studies in the laboratory.

2. For today's standards, the display of the alleles (genetic strategy) used in this study cannot occupy a full main figure (Figure 1).

Figure 1 was removed.

3. The plethysmograph traces presented in Figures 2, 4, 5 and 6 should be accompanied by a period of pre-Gas exposure and post-Gas exposure.

Pre-gas exposure and post-gas exposure was added in Figures: 3 and 4. Additionally, all figures were restructured.

4. Figure 7 and 8 could be combined, and more representative photographs should be presented. The assignment of the facial motor nucleus seems to be arbitrary, as it lacks Phox2b immunoreactive cells. Facial motor neurons do express Phox2b in addition to ChAT in the adult life of mice.

We added more representative figures, as well as control experiments. For that reason, we keep two figures for the anatomical representations. We do not have a final answer why we did not have an Phox2b-immunoreactive in the facial motor nucleus. However, previous experiments from our lab (PMID: 22015927, 24286756, 24363384, 27633663) and from different labs (PMID: 17021186, 17255166, 18440993, 26068853, 29066557) performed in postnatal/adult mice did not found Phox2b-expressing neurons in the facial motor nucleus. That are in contrast to what have been showed during embryonic stages.

5. The general outline of the manuscript is nothing I have seen before in eLife, that is numbered points for Materials and methods and the result section, although I admit that it helped a lot with the reading.

Thank you for the positive comments.

6. The authors show a reduction (about half) in the number of Phox2b+/TH- cells in adult Atoh1Cre,Phox2bdelta8 mice, and assume that this is indicative of a reduction in the number of retrotrapezoid neurons. This is not necessarily true. I would recommend that the authors present first an anatomical/developmental characterization of retrotrapezoid neurons in Atoh1Cre,Phox2bdelta8 at embryonic and neonatal states (E12.5, E16.5 and P0). At this stages, retrotrapezoid neurons have a well-established molecular signature: Phox2b, Atoh1 and Lbx1 expression. Whereas there are not great commercial Lbx1 and Atoh1 antibodies, the authors could consider combining in situ hybridization for Lbx1 and/or Atoh1 with Phox2b immunoreactivity, the Phox2b antibody that the authors used in this study is great, and compatible with in situ hybridization (my own experience). Other studies by the groups of Huda Zoghbi and Jean-Francois Brunet have shown that interfering with Phox2b and Atoh1 expression in retrotrapezoid neurons results in the incorrect location of this cells dorsally to the facial motor nucleus, is this phenotype also present in Atoh1Cre,Phox2bdelta8 mice?

The Phox2b+/TH- staining to characterize chemosensitive neurons in the RTN/parafacial region has been extensively demonstrated (PMID: 26068853; 29066557). Additionally, in vitro, and anatomical data showed that Phox2b+/TH- neurons in the RTN responded to CO_2_/H^+^ and majority express TASK-2 and GPR4 receptor. Respectively, a H^+^inhibited background K^+^ channel, and a H^+^-activated G-protein-coupled receptor, that strongly suggest mediating the chemosensitivity of RTN/parafacial neurons. Thus, it is unlikely that the population showed in our study did not involve chemosensitive neurons in the RTN/parafacial region. In addition, we used a second strategy, Fos+/TH- immunostaining, to investigate the RTN/parafacial neuronal activation induced by hypercapnia and to avoid the C1 adjacent neurons. Fos+/TH- neurons expression reduced by 50% in adult mutated mice compared to control. Together, these results indicate that the reduction in Phox2b+/TH- neurons in the mutant mice greatly reduced the number of functional chemosensitive neurons from RTN/parafacial region.

We understand the importance to describe the anatomical characterization of RTN/parafacial neurons at embryonic stages, but it was not the main goal of the present study. We did not analyze any functional respiratory parameters during embryonic stages, as most studies have already done. We focus in characterize functional and anatomical changes induced by the NPARM Phox2b^Δ8^ mutation in a mature system. As it corresponds with the timeline when CCHS is detected, to investigate how the anatomical changes could correlate with functional changes when the mutation is activated at late embryonic stages in a restricted neuronal population (periVII and periV). From now, to better understand the mechanisms we will consider the suggestions in the future studies.

The conditional NPARM Phox2b^Δ8^ in Atoh1-expressing cells did not result in miss location of Phox2b neurons dorsally to facial motor nucleus. Our study used a similar strategy that Ruffault et al. (2015) used in their study. The difference is that instead activated the NPARM Phox2bΔ8, they deleted Phox2b from Atoh1-expressing cells (Atoh1^Cre^;Phox2b^lox/lox^). Interestingly, improper location seems to depend directly on Atoh1. Since, deletion of Atoh1 from same population (P2b::CreBAC1;Atoh1lox/lox) shifted Phox2b neurons dorsally. In addition, similar results occurred when Atoh1 was deleted within the HoxA4 domain (PMID: 22958821).

7. The authors show a reduction of Fos+/TH- cells in adult Atoh1Cre,Phox2bdelta8 mice that were exposed to high levels of CO2 in air. From this result the authors conclude that the number of Fos-activated retrotrapezoid neurons is decreased. Again, this is not necessarily true. To better define this, it is necessary to demonstrate that Fos is not express in Phox2b+/TH- retrotrapezoid neurons. Whereas the Fos and Phox2b antibodies used in this study are both generated in rabbits, the authors could make use of the eGFP expression present in the Phox2bdelta8 allele (Figure 1), therefore, it is necessary to combine the immunoreactivity for eGFP (Phox2b), Fos and TH. This is central for this study, as if indeed less retrotrapezoid neurons are activated by CO2 in adult Atoh1Cre,Phox2bdelta8 mice, it is astonishing that these mice can have a full response to hypercapnia. This is intriguing because other mouse models, in which the number of retrotrapezoid neurons are reduced in greater numbers, do not show a full response to CO2 in the adult life, for instance in: P2b::CreBAC1;Atoh1lox/lox (Ruffault et al., 2015), Egr2cre;P2b27Alacki (Ramanantsoa et al., 2011, DOI: 10.1523/JNEUROSCI.1721-11.2011), Atoh1Phox2bCKO mice (Huang et al., 2017, DOI: 10.1016/j.neuron.2012.06.027) and Egr2cre;Lbx1FS (Hernandez-Miranda et al., 2018, DOI: 10.1073/pnas.1813520115).

The Phox2b and fos antibody that we currently used in the laboratory are both made in the same species (rabbit). Thus, to further avoid any misinterpretation by counting C1 adjacent neurons that also could be activated by hypercapnia. We applied the absence of TH- immunostaining together with anatomical well-defined region (ventral and laterally to facial nucleus) as a criterion to define RTN/parafacial neurons. We also added more information to the manuscript to be clear to the reader. Regarding the use of GFP, in concordance with our prior data (Nobuta et al., Alzate et al.), GFP expression is only detected in embryonic Phox2b expressing cells. Although the most likely reason for this is NPARM Phox2b induced lethality, we acknowledge that other possibilities such as incomplete cre-mediated recombination or Phox2b mutation self-repression may mediate these effects (PMID: 2310352; 17765533).

As far as the above studies mentioned by the reviewer, the RTN/parafacial neurons investigated were done only during embryonic stages. Therefore, we did not know if in adult, the number of neurons would be comparable to the embryonic age showed to further discusses with the data presented by our study. In addition, the former study, they did not perform any experiment to investigate whether the remaining parafacial/RTN neurons were functional or not.

8. The authors do not address if retrotrapezoid neurons/parafacial cells are rhythmically active and responsive for pH changes in the embryonic/neonatal life of Atoh1Cre,Phox2bdelta8 animals. The fact that these mice can fully respond to hypercapnia in the adult life but not in the neonatal stages might imply that retrotrapezoid neurons are present but somehow silenced in Atoh1Cre,Phox2bdelta8 neonates. Therefore the anatomical/developmental characterization of retrotrapezoid neurons (as suggested above) should be complemented with in vitro calcium imaging in Atoh1Cre,Phox2bdelta8 embryos or neonates. This could explain why mice that completely lack retrotrapezoid neurons (Egr2cre;Lbx1FS) do not fully display the hypercapnic reflex, whereas Atoh1Cre,Phox2bdelta8 mice do.

Thank you for your comments, which are essential to complement our study. However, in our laboratory, we do not have the option of carrying out calcium imaging experiments. Future projects and experiments and maybe some collaborations are being proposed, so that we can answer this question, as well as others.